

# Atlantic meridional overturning circulation at 14.5° N and 24.5° N during 1989/1992 and 2013/2015: volume, heat and freshwater fluxes

Yao Fu[1], Johannes Karstensen[1], Peter Brandt[1, 2]

[1]GEOMAR Helmholtz Centre for Ocean Research Kiel, Kiel, Germany

[2]Christian-Albrechts-Universität zu Kiel, Kiel, Germany

*Correspondence to*: Yao Fu (yfu@geomar.de)

**Abstract.** The Atlantic meridional overturning circulation (AMOC) is analyzed using hydrographic data from trans-Atlantic sections along 14.5° N, occupied in 1989 and 2013, and along 24.5° N, occupied in 1992 and 2015. Comparison between the periods shows that the Antarctic Intermediate Water (AAIW) became warmer and saltier at 14.5° N, and the density of the

densest Antarctic Bottom Water decreased at both sections. By applying a box inverse model, the absolute meridional velocity across the sections and dianeutral velocity across neutral surfaces were determined. Corresponding to the warming and salinification of the AAIW at 14.5° N, the intermediate layer transport was also considerably weaker in 2013 than in 1989. The AMOC was generally stronger during 1989/1992 than during 2013/2015 (18.6±2.7 vs. 14.7±3.9 Sv at 14.5° N, and 19.2±1.7 vs. 16.9±1.6 Sv at 24.5° N, respectively; 1 Sv=$10^6$ m$^3$ s$^{-1}$). The inverse solution suggests that the transport

changes are caused by reduction in the northward thermocline and intermediate water transport, which is balanced by decrease in the southward upper North Atlantic Deep Water transport at both sections. The AMOC strength of the inverse solution agrees well with that of dynamically consistent and data-constrained ocean state estimate GECCO2 (15.8±3.4 Sv at 14.5° N, and 17.7±3.6 Sv at 24.5° N) and derived by the RAPID array data (16.9±4.4 Sv), but is generally smaller than that derived by the MOVE array data (24.1±4.1 Sv). Instead of any long-term trend, the GECCO2 shows strong seasonal to

interannual variability of the AMOC at both latitudes, which may explain the observed changes of the AMOC in the box inverse model. Sensitivity tests of the inverse solution suggest that the overturning structure and heat flux across the 14.5° N section are sensitive to the Ekman transport, while freshwater flux is sensitive to the transport-weighted salinity at the western boundary.

## 1. Introduction

The Atlantic Meridional Overturning Circulation (AMOC) plays an important role in the global climate. In the tropical and subtropical North Atlantic, it consists of northward flowing surface, intermediate, and bottom waters, and southward flowing North Atlantic Deep Water (NADW). The warm and shallow net northward flow carries a large amount of heat to the subpolar North Atlantic, where the heat is released to the atmosphere, resulting in the formation of NADW. The NADW then





returns as a cold and deep southward flow mainly within the deep western boundary current (DWBC). The net heat flux from the tropics to the higher latitudes has large climate impact (Wunsch, 2005).

Since decades, oceanographers have continuously spent efforts investigating different aspects of the AMOC. For instance, using hydrographic section data obtained during the International Geophysical Year (IGY), and the World Ocean Circulation

Experiment (WOCE), many studies focused on estimating the volume transport, and heat and freshwater fluxes related to the AMOC (e.g. Roemmich and Wunsch, 1985; Macdonald, 1998; Ganachaud and Wunsch, 2000, 2003; Lumpkin and Speer, 2003). These pioneer studies revealed a consistent picture of the AMOC in terms of large-scale horizontal and overturning circulation. Recent studies based on end-point geostrophic mooring data (e.g. Meridional Overturning Variability Experiment, MOVE; and Rapid Climate Change-Meridional Overturning Circulation and Heat Flux Array-Western

Boundary Time Series, RAPID-MOCHA-WBTS), have shown that the AMOC exhibits variability on timescales from seasonal to decadal (Cunningham et al., 2007; Kanzow et al., 2008, 2010; Smeed et al., 2014). At 26° N, different components of the AMOC have been analyzed in details including the western boundary current (Florida Current), the Ekman transport, and the interior geostrophic transport. Based on RAPID-MOCHA-WBTS (hereafter RAPID array), Cunningham et al. (2007) reported that on timescales from subseasonal to seasonal, the upper ocean has an immediate

response to the change in the Ekman transport, while Kanzow et al. (2010) and McCarthy et al. (2012) have shown that on the seasonal to interannual timescales, the interior geostrophic transport plays a more important role in modulating the strength of the AMOC. Interestingly, studies based on the RAPID array at 26° N and on the MOVE array at 16° N often reveal contrary conclusions on the AMOC. For instance, RAPID-based estimates indicate that the AMOC decreased at a rate of $-4.1 \pm 3.2$ Sv decade$^{-1}$ between Apr 2004 and Oct 2013, while MOVE-based estimates indicate a strenghening trend of

$8.4 \pm 5.6$ Sv decade$^{-1}$ during the same period (Baringer et al., 2015). Smeed et al. (2014) found that a strengthened southward thermocline water transport is responsible for the weakened AMOC at 26° N during 2004-2012, which was balanced by a decrease in the southward flow of the lower NADW. However, Send et al. (2011) found that the interannual variability in the Labrador Sea Water (LSW; as the primary component of the upper NADW) is responsible for the interannual variability in the AMOC at 16° N.

A trans-Atlantic section nominally along 14.5° N was occupied twice in the past, in February-March, 1989, and in May-June, 2013 (Fig. 1). One of the unique feature of this section is that it lays roughly at the latitude of maximum annual mean northward meridional Ekman transport (about 8.3 Sv, 1 Sv $= 10^6$ m$^3$ s$^{-1}$) in the tropical North Atlantic (Fu et al., 2017). Therefore, it is expected that at this latitude, the meridional Ekman transport plays a more important role in the AMOC than at mid-latitudes, for instance, 26° N, especially in terms of meridional heat flux associated with the AMOC.

A classical method of estimating the AMOC is the application of a box inverse model to an oceanic box bounded by hydrographic sections and landmasses. By conserving the volume (or mass) and other properties (e.g. heat and salt) in the whole box and layers, the absolute geostrophic velocity (thus the absolute transport) can be obtained (Wunsch, 1977, 1996). In this study, we apply a box inverse model to realizations of the 14.5° N (1989 and 2013) and 24.5° N (1992 and 2015) sections (Fig. 1). The results of the box inverse models for the two different periods will be presented and discussed. The





24.5° N section, aligned with the RAPID array, is one of the most frequently surveyed basin-wide zonal sections in the North Atlantic. Many aspects of the AMOC along 24.5° N, in terms of transport components, circulation patterns, and heat and freshwater fluxes, are well described, and provide reliable information on constraining a box inverse model (Roemmich and Wunsch, 1985; Ganachaud, 2003; Lumpkin and Speer, 2003; Atkinson et al., 2012; Hernández-Guerra et al., 2014).

Therefore, we chose this section in combination with the 14.5° N section for the box inverse model, which allow us to pay more attention to the less well studied features associated with the 14.5° N section.

Klein et al. (1995) calculated the meridional transport at 14.5° N from the 1989 realization by applying a box inverse model in combination with the 8° N section occupied in 1957. When compared to modern data (Fu et al., 2017), Klein et al. (1995) used an extremely large annual and seasonal (winter) Ekman transport of 13.58 Sv and 15.94 Sv for the calculation, in

contrast to an annual mean of 8.29±2.59 Sv calculated from the National Centers for Environmental Prediction Climate Forcast System Reanalysis (NCEP/CFSR) monthly wind stress data. Besides, the large time difference between the occupations of the two sections (more than 30 years) could be problematic in the application of a box inverse model. As a result, the derived transport of northward flowing water masses (e.g. Antarctic Intermediate Water, AAIW and Antarctic Bottom Water, AABW) was surprisingly small. Therefore, it is also necessary to revisit the 1989 realization of 14.5° N in

order to obtain a more reliable meridional overturning circulation for that year.

The paper is organized as follows: In section 2, the data used in this study are introduced, and the water mass distribution and the property changes of the water masses between the two periods along the two latitudes are described. The details of the box inverse model and the calculation methods for the heat and freshwater fluxes are given in section 3. The inverse model results are presented and discussed in section 4. A comparison between the inverse solutions, GECCO2 ocean state

estimate, and the MOVE and RAPID array analysis data is presented in section 5. A conclusion is given as the final section.

## 2 Hydrographic data and water masses

### 2.1 Hydrographic sections at 14.5° N and 24.5° N

The first survey of the 14.5° N section was from 5 to 23 February 1989, and jointly completed by R/V Meteor (M09) and R/V Albatross IV, covering the eastern and western half of the section separated by the MAR (at about 42.5° W),

respectively. At the western boundary, the cruise track of R/V Albatross was northeastward from 11.3° N/60.3° W to 14.5° N/ 57.5° W. It was necessary in order to complete the section perpendicular to the continental slope and to sample the western boundary currents (including the DWBC) there. During the cruise, Conductivity-Temperature-Depth (CTD) measurements were conducted. At the western boundary, all of the CTD profiles (westernmost 10 profiles) reached the bottom with an averaged horizontal spacing of 30 nm, for the rest of the section, the CTD depth alternated between the

bottom and 2000 m with an averaged horizontal spacing of 45 nm. As a result, 39 CTD profiles in total reached the bottom for the entire section with a 1-m vertical resolution. The accuracy of the data set is expected to be ±0.002 °C for temperature, ±0.002 psu for salinity, and ±3 dbar for pressure, respectively (Klein et al., 1995). The M09 CTD was also equipped with a




Beckmann type polarographic oxygen sensor that measured the oxygen current (OC/A) and oxygen temperature (OT/°C). They were converted to dissolved oxygen concentration according to Owens and Millard (1985) and calibrated using the available bottle oxygen data. The final processed CTD oxygen data have an accuracy of ±10 µmol kg$^{-1}$ (using a double root mean squared difference against bottle oxygen data). The R/V Albatross CTD oxygen sensor malfunctioned; only bottle

oxygen data were available. The processing details of the R/V Albatross and R/V Meteor CTD data have been described in Wilburn et al. (1990) and Klein (1992).

The second survey of the 14.5° N section was done in 2013 aboard R/V Meteor during the cruise M96 (28 April to 20 May 2013) and M97 (8 and 9 June 2013). The M96 leg covered the section from the coast of Trinidad and Tobago (11.3° N/60.3° W) to about 20° W, and the M97 leg covered from 20° W to the African coast (easternmost 4 stations). During the cruises,

CTD measurements were performed on a grid similar to that of 1989. The westernmost part of the section (10 stations) was perpendicular to the coast and all profiles reached the bottom; for the rest of the section, CTD depth alternated between the bottom and 3000 m. In total 64 CTD stations were occupied along the 14.5° N section with a nominal resolution of 40 nm, and 48 profiles reached the bottom with a 1-m vertical resolution. The CTD work was carried out using a Sea-Bird Electronic (SBE) 9 plus system with double sensor packages for temperature, conductivity and oxygen. The accuracy of the

temperature sensors is ±0.001 °C; after calibration by comparing the bottle stop salinity and oxygen with the measurements of salinometer and oxygen titration of bottle samples, the accuracy of salinity and oxygen sensors is ±0.002 psu and ±1.3 µmol kg$^{-1}$, respectively. Different from the 1989 occupation, the 2013 occupation also included direct velocity observations. During the M96 leg, two 300 kHz Teledyne RDI Lowered-Acoustic-Doppler-Current-Profilers (LADCP) were mounted to the CTD-Rosette for most of the stations, except for stations between 38° W and 33.5° W, where the large water

depth (>6000 m) prevented the use of the LADCPs. During the M97 leg, no LADCP system was installed. The LADCP data was post-processed with the GEOMAR processing routine v10.4 (Fischer and Visbeck, 1993; Visbeck, 2002). Additionally, two vessel-mounted Teledyne RDI ADCP (SADCP) were in operation at frequency of 38 kHz and 75 kHz. They are equipped with phased array transducers and can measure up to 1200 m and 800 m depth, respectively. Ship navigation information was synchronized with the system. Misalignment angles and amplitude factors were calibrated during the post

processing. The uncertainties of 1-hour averages of the underway measurement is expected to be 2-4 cm s$^{-1}$ (Fischer et al., 2003).

At 24.5° N, the CTD section has been repeated several times (Atkinson et al., 2012). In this study, we chose data measured during cruises from 14 July to 15 August 1992, and from 10 Dec 2015 to 20 Jan 2016 to combine with the 14.5° N section measured in 1989 and 2013, respectively. The CTD data were provided through the CLIVAR and Carbon Hydrographic

Data Office (CCHDO). All the profiles reached tens of meters above the ocean bottom with a 2-m vertical resolution. The distance between the adjacent CTD stations was typically 25 to 30 nm. When surveying across boundary currents and continental slopes, the distance was reduced to resolve the boundary current structure. Within the Florida Strait, 11 profiles were collected as part of the 1992 section. During the 2015 occupation, Florida Strait was surveyed twice, but only profiles collected during the first survey were used in this study due to the higher spatial sampling resolution.





### 2.2 Water mass distribution and property changes

The water mass properties at 14.5° N have been introduced by Molinari et al. (1992) and Klein et al. (1995) based on the 1989 realization. At 24.5° N, the water mass properties have been described several times (e.g. Roemmich and Wunsch, 1985; Hernández-Guerra et al., 2014). Here we briefly describe the water mass properties, and emphasize the differences

between the former and latter periods at each section. The potential temperature ($\theta$) and salinity ($S$) diagram drawn using data at the two sections provides an overview of the distribution of water masses (Fig. 2). The vertical sections of salinity, potential temperature, oxygen ($O$), and neutral density ($\gamma_n$), are presented for the different years at each section in Fig. 3-6. As described in the previous section, at 14.5° N, except in the western boundary region, the CTD depth alternates between 2000 m (3000 m) and the ocean bottom for the measurement in 1992 (2013). For a better visual effect, a Gaussian weighting

function with horizontal influence and cut-off radii of 0.5 and 1.5 degree in longitude, vertical influence and cut-off radii of 20 and 40 m were applied to interpolate and smooth the data. The horizontal and vertical resolution of the mapping was chosen as 0.25 degree and 20 m. At 24.5° N, no interpolation and smoothing are needed due to the high horizontal resolution. The difference of $S$ and $\theta$ between the two time periods for each section is also shown to facilitate the discussion (Fig. 3c and 4c). To calculate the difference, the original $S$ and $\theta$ were firstly transformed from pressure levels to uniform $\gamma_n$

levels, then they were horizontally interpolated on each $\gamma_n$ level onto a consistent horizontal grid; finally, the differences of $S$ and $\theta$ were calculated between the two time periods, and transformed back to pressure level and smoothed using a Gaussian weighting function as described above.

At 14.5° N, only bottle oxygen data are available in the western half of the basin for the 1989 dataset (the Albatross IV leg). To obtain an oxygen section on the same regular grid, the bottle oxygen, $\theta$ and $S$ data were used to derive oxygen as a

function of $\theta$ and $S$ in the $\theta$/$S$ space. Here a Gaussian interpolation function was applied to achieve a general functional relationship that finally allowed obtaining an oxygen distribution from the $\theta$ and $S$ distribution along the section. To further correct for local variations that may not have been correctly represented by the functional relationship, we calculated an oxygen anomaly as the difference between derived oxygen and bottle oxygen data. The oxygen anomaly was interpolated onto the grid of the section and finally subtracted from the derived oxygen field. In this way, both general and local relations

of the oxygen to the water masses were taken into account. The remaining eastern part of the 1989 oxygen section is directly from the CTD Beckman type oxygen sensor (cf. Section 2.1).

The water mass distribution can be inferred from the $\theta$-$S$ diagram (Fig. 2). At 14.5° N, proceeding from the surface, the long tail with homogeneous temperatures above 25 °C represents the surface water mainly in the mixed layer. The different temperature levels (with a mean offset of 1.60 °C in the upper 50 m) are most likely caused by the different seasons when the

occupations took place (February 1989 and May 2013, respectively). Slightly below the surface water, a salinity maximum at about 24 °C marks the Subtropical Underwater (STUW), corresponding to the subsurface layer centered at about 120 m in Fig. 3a and 3c. This water mass originates from the subtropical Atlantic, where evaporation exceeds precipitation, forming sea surface salinity (SSS) maxima at about 25° N in the North Atlantic (as shown in Fig. 3b and 3d at 24.5° N) and about 20°



S in the South Atlantic. The SSS maxima are further subducted and transported equatorward and westward as part of the subtropical cells (McCreary and Lu, 1994; Schott et al., 2004). The median $\theta$-$S$ diagram calculated on neutral density surfaces at 14.5° N shows two salinity maxima between 22 and 25 °C (Fig. 2a), indicating the presence of northern and southern origin of these water masses (Klein et al. 1995).

Within the temperature range from 20 to 8 °C, the majority of water has a nearly linear $\theta$-$S$ relationship for both sections (Fig. 2) that is a characteristic of Central Water (CW). CWs compose the main thermocline and are formed by subduction in the subtropical gyre. As a result of subtropical gyre circulation, CWs occupy a larger depth range in the west than in the east, which can be seen from the upward tilting of the isotherms toward east (Fig. 4a to 4d). Within the temperature range between 13 and 18 °C, a deviation from the linear $\theta$-$S$ relation is found at the 14.5° N section from profiles sampled east of

22° W, where the Cape Verde Frontal Zone (CVFZ) is located. East of the CVFZ, signatures from South Atlantic Central Water (SACW) are seen, which is less saline than North Atlantic Central Water (NACW). The SACW takes routes along and parallel to the equator following, after crossing the equator in the western equatorial Atlantic, the different eastward jets of the tropical North Atlantic. When approaching the West African coast, it spreads further northward from where it diffuses into the eastern tropical North Atlantic (Kirchner et al., 2009; Brandt et al., 2015). The CW layer at 24.5° N consists

primarily of NACW.

At 14.5° N, the CW layer is characterized by low dissolved oxygen (DO) concentration down to 40 µmol kg$^{-1}$ at its core (Fig. 5a, c). This layer is within the Oxygen Minimum Zone (OMZ) in the eastern tropical North Atlantic, which is formed generally due to weak ventilation and oxygen consumption (Luyten et al., 1983; Brandt et al., 2015). Comparing the OMZ between 2013 and 1989, it shows that the thickness of the minimum oxygen core (DO<60 µmol kg$^{-1}$) increased, while the

westward extent of this core reduced. Such changes created a complicated pattern of the DO difference between 2013 and 1989 (not shown). First of all, the vertical expansion of the minimum oxygen levels are in agreement with the findings by Stramma et al. (2010), and created a local decrease of DO below the near-surface layer down to 500 m along the eastern boundary (west of 22° W). On the other hand, the reduction of its zonal extent led to an increase of DO to the east of the core position in 2013, indicating that the oxygen decrease is not spatially coherent with local processes, and/or that processes on

shorter timescales might affect the oxygen evolution.

Directly below the CW is the intermediate water layer. At 14.5° N this layer consists predominantly of AAIW, while at 24.5° N AAIW and Mediterranean Water (MW) are both main contributors to this layer. AAIW is characterized by its salinity minimum centered at around 900 m (Fig. 3a and 3c), while MW is warmer and much saltier. The AAIW in the North Atlantic originates from the circumpolar South Atlantic, and is transported northward mainly along the western boundary

(Tsuchiya, 1989). As a consequence of mixing with the surrounding waters on the way toward the north, AAIW gradually loses its salinity minimum signature in the northern hemisphere from the tropics to the subtropics. This can also be seen when comparing the 14.5° N and 24.5° N sections.

From the $\theta$-$S$ diagram (Fig. 2a, c) and the vertical sections (Fig. 3e and 4e), it can be seen that the AAIW at 14.5° N became saltier and warmer between 1989 and 2013. This result is consistent with the previous studies on AAIW by Sarafanov et al.





(2008), who found a significant warming of the AAIW at 6.5° N between 1993 and 2000. Schmidtko and Johnson (2012) reported a warming and salinification trend in the AAIW core since 1970s throughout the tropical North Atlantic. They have attributed the cause of the warming trend to higher sea surface temperature in the AAIW formation region in the background of global warming, and to a strengthened Agulhas leakage associated with a low SAM during some periods of the 20th

century. At 24.5° N, between 2015 and 1992, the waters in the intermediate layer became generally cooler and fresher in the western half of the section, while warmer and saltier in the eastern half of the section (Fig. 3f and 4f). Since the intermediate water at 24.5° N is composed of both AAIW and MW, the property change over time could be considered as an imprint of changing relative contributions of AAIW and MW along the section. On the basin scale, a strengthening of the gyre scale circulation would bring more AAIW on the western side from the tropics, while also more MW on the eastern side from the

north, which would create a pattern like the one we observed. Locally, for instance along the eastern margin, higher salinity in the 2015 realization than in the 1992 realization may reflect a seasonal intrusion of the less saline water near the African coast (Roemmich and Wunsch, 1985; Hernández-Guerra et al., 2005). MW has been reported to occupy a greater portion of the eastern boundary during late winter and early spring, while AAIW contributes more in late fall and early winter (Hernández-Guerra et al., 2003; Fraile-Nuez et al., 2010). Machín et al. (2009, 2010) explained the seasonally varying

contribution of the AAIW and MW with the stretching and shrinking of the intermediate water strata. Mediterranean eddies appeared at 24.5° N in both realizations and caused the maximum salinity values at about 7 °C in the $\theta$-$S$ diagram (Fig. 2b). The NADW originates from the subpolar North Atlantic and is found at both sections (14.5° N and 24.5° N) in the $\gamma_n$ ranges of 27.922–28.1295 kg m$^{-3}$. According to its formation region and the corresponding density ranges, NADW can be divided into Upper NADW (UNADW), which is composed primarily of water formed in the Labrador Sea (LSW) (Talley and

McCartney, 1982), and Lower NADW (LNADW), originating from water masses formed in the Nordic Sea (Pickart, 1992; Smethie et al., 2000), namely Iceland-Scotland Overflow Water (ISOW) and Denmark Strait Overflow Water (DSOW). At both sections, a sharp isoneutral (surface of the same neutral density) slope in the NADW layers at the western boundary is indicative of the DWBC and its recirculation. The DWBC seems to be a continuous feature between 24.5° N and 14.5° N, which is evident from the elevated oxygen concentration (DO>260 μmol kg$^{-1}$) along the continental slope at the western

boundary (Fig. 5) due to a more recent contact with the atmosphere compared with surrounding waters in a similar density range. A dual-core structure of high oxygen at the western boundary is captured at 14.5° N in 2013, resulting from the ISOW with relatively low oxygen laying between the oxygen-rich LSW and DSOW. At 24.5° N, even higher oxygen concentration (DO>270 μmol kg$^{-1}$) is found in the DWBC region centered at about 3300 m in 2015, indicating the presence of more recently ventilated contributions to the LNADW. Throughout the NADW layers, the water became fresher and cooler on

neutral density surfaces at both sections. The freshening is in agreement with previous studies showing that the NADWs in the formation region has been freshening since decades (Dickson et al., 2002).

Isotherm and isohaline surfaces do not continue from the western into the eastern Atlantic across the Mid-Atlantic-Ridge (MAR) below 3500 m at both latitudes. Lower temperature and salinity but higher density is found in the western basin (west of the MAR) due to the influence of less saline and colder but also denser AABW (e.g. Klein et al., 1995). AABW is




the densest water in the world oceans and is present at both 14.5° N and 24.5° N primarily in the western basin with $\gamma_n$ larger than 28.1250 kg m$^{-3}$ and $\theta$ lower than 1.80 °C. The MAR, as a topographic barrier, prevents AABW flowing directly into the eastern Atlantic basin, except at the Romanche Fracture Zone near the Equator and the Vema Fracture Zone at 11° N (Wüst, 1935; McCartney et al., 1991). Klein et al. (1995) reported that the lowest salinity and temperature in the eastern basin found

at 14.5° N was lower than that found at either 16° N and 8° N, indicating that a pathway for AABW crossing the MAR existed between 8° N and 14.5° N. It is believed that AABW gets warmer and saltier on its way to the north due to mixing with the overlying LNADW, therefore, it is not surprising that the densest AABW is warmer and saltier at 24.5° N than at 14.5° N.

It appears that at both latitudes, the AABW became cooler and fresher between the two periods (Fig. 3e, f and 4e, f). Note

that we have calculated the changes on neutral surfaces, if the difference is calculated on pressure surfaces, it presents a different and complex picture (not shown). For example, at 14.5° N, the pressure-based difference shows that the AABW became warmer and salter west of 55° W, while cooler and fresher east of 55° W. At 24.5° N, the pressure-based difference shows that the AABW became mostly warmer but fresher, which is consistent with the previously observed warming of AABW at 24.5° N by Johnson et al. (2008), who also calculated the changes on pressure surfaces. At the same time, we

noticed that the density of densest AABW observed at 14.5° N reduced from 28.1686 kg m$^{-3}$ in 1989 to 28.1623 kg m$^{-3}$ in 2013, and at 24.5° N from 28.1596 kg m$^{-3}$ in 1992 to 28.154 kg m$^{-3}$ in 2015. As a result, we believe that the warming of the AABW at 14.5° N and 24.5° N should not arise from property changes on the density surface, but should be due to depletion of the densest AABW core ($\gamma_n > 28.141$ kg m$^{-3}$) at both latitudes. This agrees with Herrford et al. (2017), who showed that in the equatorial South Atlantic, the coldest AABW became warmer since the early 1990s.

## 3 Methods

### 3.1 Inverse model setup

The box inverse method was applied to determine the meridional overturning transport across the 14.5° N and 24.5° N sections. Two "boxes" bounded by the ocean boundaries and the hydrographic sections measured in 1989/1992 (1989/1992 box) and 2013/2015 (2013/2015 box), respectively, can be built. Based on the thermal wind relation, the vertical shear of the

geostrophic velocity relative to an arbitrary reference depth can be calculated from the density field between adjacent CTD stations. To obtain the absolute geostrophic velocity, a reference velocity must be assigned to the shear at the reference level. A box inverse model is composed of equations that employ the thermal wind relation and conservation of properties (i.e. volume, salt, heat, Eq. 1). It adjusts the reference geostrophic velocity and vertical fluxes of properties from prior (initial) estimates to conserve the properties within the whole box and each layer of the box.

Following Ganachaud (2003) and Hernández-Guerra et al. (2014), we separated the two boxes into 17 vertical layers by neutral density surfaces specified according to the characteristics of water masses (Table 1). In this study, we applied the conservation of volume, salt anomaly, and heat at each layer and the whole box. The salt anomaly was calculated by



subtracting a section-areal mean salinity from the salinity values. The conservation equation of a property $C$ in a layer can be formulated in the general form

$$\sum_{j=1}^{J} \Delta x_j \int_{h_m}^{h_{m+1}} (v_j + b_j) C_j \, dz + [-w_m^C A_m C_m + w_{m+1}^C A_{m+1} C_{m+1}] \approx 0 \,, \tag{1}$$

where $j$ and $m$ are indices for station pairs and layer interfaces, respectively; $J$ is the total number of station pairs; $\Delta x_j$

represents the horizontal distance between station pair $j$; $h_m$ refers to the depth of layer interface $m$; $v_j$ and $b_j$ are the relative and reference geostrophic velocities at station pair $j$, respectively; $C_j$ and $C_m$ are the property concentration at station pair $j$ and interface $m$, respectively; $w^C$ is the dianeutral velocity of property $C$ at the layer interfaces; $A$ refers to the horizontal area of the layer interfaces within the boxes. Eq. 1 states that the total change of property $C$ in one layer due to horizontal advections through the side boundaries and vertical fluxes through the upper and lower boundaries is approximately 0. Note

that we defined a dianeutral velocity for each property including the volume, salt anomaly, and heat. This has been proven to be an effective way of parameterizing the net dianeutral fluxes in inverse models (McIntosh and Rintoul, 1997; Sloyan and Rintoul, 2001; Tsubouchi et al., 2012).

Except $b$ and $w$, all other quantities in Eq. 1 are either known, or can be directly estimated from the hydrographic and topographic data. Note that for both realizations at 14.5° N, only profiles approaching to the bottom were used to construct

the box inverse model (see Sect. 2.1 for details). For the 1989/1992 box, there are 149 CTD station pairs, the adjustment to the reference velocity at each station pair is treated as an unknown; the dianeutral velocity for each property across each layer interface is treated as an unknown, which adds 51 unknowns to the system. For the 2013/2015 box, there are 228 unknowns in total, including the adjustment to the reference velocity of 177 CTD station pairs and 51 dianeutral velocities.

### 3.2 Constraints

To seek solutions for the unknowns $b$ and $w$, other transport constraints at specific locations and depths can be applied in addition to the conservation equations of the properties in each layer and the whole box (Eq. 1). The details of the additional constraints are described here and summarized in Table 2. At 24.5° N, following Hernández-Guerra et al. (2014), we assumed a DWBC transport of -26.5±13.5 Sv ("-" denotes southward). The uncertainty range was given by half of the DWBC transport, when the large uncertainty in the different estimates based on moored observations was taken into account

(Bryden et al., 2005; Johns et al., 2008; Meinen et al., 2013). This value was then applied to both time periods as the a priori constraint of the DWBC at 24.5° N between layer 7 and 14, 77° W and 72° W.

A continuous time series of the Florida Current (FC) transport has been constructed by using a succession of the submerged telephone cables across the Florida Strait between 26° N and 27° N (Baringer and Larsen, 2001; Meinen et al., 2010). In this study, the FC transport was constrained using the annual mean value of 31.3±1.1 Sv in 1992 and 31.7±1.1 in 2015,

calculated from the daily FC transport data. The uncertainties were given by the standard deviation of the annual mean FC transport between 1982 and 2016 estimated using the daily data. Note that both values are not significantly different from a





long-term mean FC transport of 31.1±2.4 Sv (adopted from Atkinson et al., 2010). Using the long-term mean value as the constraint for both boxes would only alter the final solutions marginally.

The surface freshwater flux was constrained as 0.34±0.28 Sv for both boxes (positive denotes freshwater loss of the ocean). This value was estimated using the mean and standard deviation of the monthly freshwater flux from the Hamburg Ocean

Atmosphere Parameters and Fluxes from Satellite (HOAPS-3.3) dataset between 1989 and 2013 (Kinzel et al., 2016).

About 0.8±0.6 Sv water flows from the Pacific to the Atlantic through the Bering Strait (Roach et al., 1995; Woodgate et al., 2005). In previous inverse studies involving two zonal trans-Atlantic sections, the Bering Strait transport was often regarded as a net southward barotropic flux through both sections with equal amount of volume (Hernández-Guerra et al., 2014). However, as a result, the surface freshwater flux between the atmosphere and ocean would be inevitably ignored. As

described above, we have explicitly constrained the surface freshwater flux; therefore, it is reasonable to assume different net Bering Strait transport at the two sections. In this study, we constrained the Bering Strait transport through the 24.5° N section still as 0.8±0.6 Sv, but 0.5±0.6 Sv through 14.5° N in both boxes to account for the net surface flux to the atmosphere between the two sections.

At 14.5° N, an additional constraint on the AAIW transport of 2.8±2.1 Sv for the 2013 realization was applied. We found

that without this additional constraint, the final AAIW transport for the 2013 realization after the inversion would be southward, regardless what reference level or whether an a prior reference velocity based on the ADCP measurement was applied. This is contrary to the expected AAIW flow direction. Therefore, we constrained this quantity based on the annual mean AAIW transport from the monthly data of the dynamically consistent and data-constrained ocean state estimate GECCO2 (Wunsch and Heimbach, 2006; Köhl, 2015) at 14.5° N in 2013. The uncertainty is given by the standard deviation

of the monthly AAIW transport in GECCO2 at 14.5° N between 1985 and 2015. The AAIW in GECCO2 is defined using the same neutral density range ($27.38<\gamma_n<27.82$ kg m$^{-3}$) as for the observed hydrographic data.

The Ekman transport in the box inverse model is prescribed in the first layer of each section, which was estimated using the monthly wind stress data from the NCEP/CFSR (from 1949 to 2010) and NCEP/CFS Version 2 (NCEP/CFSv2, from 2011-present). The annual mean Ekman transport in the calendar year of the cruise was used (cf. Table 2 for the values).

Considering the above listed additional constrains and the conservation equations for the properties in each of the 17 layers and the whole box, we can formulate in total 59 equations for the 1989/1992 box, and 60 equations for the 2013/2015 box. Given the much larger number of unknowns, the box inverse model is always an underdetermined system. Conventionally, the equations can be written in a matrix form

$$Ax = b, \tag{2}$$

where $A$ is a M × N matrix that consists of areas of the station pairs in layers multiplied by the property concentration, and areas of the neutral density surfaces multiplied by the property concentration on that surface; $x$ is a N × 1 matrix of unknowns that consist of reference geostrophic velocity at the station pairs and dianeutral velocity for the properties; $b$ is a M × 1 matrix that consists of property transport due to the relative geostrophic velocity; M is the number of equations; N is the number of unknowns.



### 3.3 Reference level and a prior reference velocity

The reference level for the geostrophic velocity calculation must be chosen first. An ideal choice would be a level of no motion, but which is rarely found in the real ocean. Therefore, a density surface that separates the northward flowing water masses (i.e. AAIW and AABW) from the southward flowing water masses (i.e. NADW) is preferable for the reference level

with zero velocity as an initial guess (Klein et al., 1995; Ganachaud et al., 2000). For the 14.5° N section, the reference level for the 1989 realization was chosen following Klein et al. (1995) as the surface of $\gamma_n = 27.82$ kg m$^{-3}$, which separates the AAIW and the UNADW. For the 2013 realization along the 14.5° N section and both realizations along the 24.5° N section, the reference level was defined by $\gamma_n = 28.1295$ kg m$^{-3}$, which separates the LNADW and AABW (Ganachaud et al., 2000; Hernández-Guerra et al., 2014). The neutral surfaces with $\gamma_n \geq 28.1295$ kg m$^{-3}$ tilt upward toward the east on the western

side of the MAR. Therefore, it would favor a northward flow in the AABW layers when using $\gamma_n = 28.1295$ kg m$^{-3}$ as the reference level with zero reference velocity (Fig. 6). For all the sections, when the corresponding density level is deeper then the deepest common depth of the adjacent CTD station pairs, the deepest common depth was then chosen as the reference level. The bottom triangle was set by constantly extrapolating the velocity at the deepest common depth to the bottom. Using these reference levels with zero reference velocity for the sections results in an AMOC-like meridional transport with, for

instance, water in the NADW layers (layer 7 to 14) generally flowing southward, and in the bottom layers (layer 15 to 17) flowing northward. However, the imbalance between the two sections has the same order of magnitude as the initial transport itself, indicating that the volume is not conserved in individual layers and the whole box at this stage. Here "initial transport" corresponds to the transport derived from the geostrophic velocity assuming zero velocity at the chosen reference levels.

In the lack of a prior knowledge on the absolute velocity field, the reference velocities at the reference level are assumed to be 0. This is the case for both realizations at 24.5° N, and for the 1989 realization at 14.5° N. During the occupation along 14.5° N in 2013, both SADCP and LADCP were operated. These direct velocity observations provide valuable information about the velocity at the reference level. Dengler et al. (2002) showed that the vertically averaged LADCP velocity could serve as the reference geostrophic velocity for the corresponding CTD station pairs with an accuracy of $0.012\ m\ s^{-1}$.

Following their method, the barotropic tide component was first removed from the cross-section LADCP velocity, and then the tide-removed LADCP velocity was averaged between the adjacent CTD stations. The barotropic tide velocity was estimated by using the Tide Model Driver (TMD) (Egbert and Erofeeva, 2002). Fu et al. (2017) showed that for the 2013 realization, meridional ageostrophic velocity existed mainly in the upper 200 m. Hence, the cross-section LADCP velocity and relative geostrophic velocity were averaged vertically between 200 m and 50 m above the bottom to avoid the influence

of any ageostrophic component. Finally, the difference between the vertically averaged LADCP and relative geostrophic velocity was calculated to represent the reference geostrophic velocity. Note that at the western boundary region (westernmost 6 station pairs west of 57.5° W), cross-section velocity from the 38-kHz SADCP was used instead of LADCP velocity. Because the water depth is relatively shallow (less than 1300 m), and the 38-kHz SADCP covered most of the





water column with a much higher horizontal resolution (10-minute ensembles), which makes the horizontal average of the SADCP velocity profiles between the CTD stations more preferable than the two-profile average of LADCP velocities. As shown in Fig. 7, the reference velocity for the 2013 realization at 14.5° N shows large positive values at the western boundary, indicating a boundary intensified current. The error bars for station pairs 1-6 were estimated from the root mean

squared error of the vertically averaged SADCP velocity (in the range of 200 m and 50 m above the last reliable bin) between each station pairs; for the rest of the station pairs, the uncertainties were given as 0.04 m s$^{-1}$ in the western basin and 0.02 m s$^{-1}$ in the eastern basin, which are larger than the value estimated by Dengler et al. (2002), but is generally assumed for the reference velocity in box inverse model studies (e.g. Ganachaud, 2003).

### 3.4 Weighting and error estimates

The solution of the box inverse model depends on the a priori knowledge about transport variation as well as uncertainties in the reference velocity field. This can be translated into a row weighting matrix $W$ (M × M) and a column weighting matrix $E$ (N × N), which were applied to the system before inversion. Following Tsubouchi et al. (2012), the row weighting for volume conservation is defined as

$$W_{mm} = \frac{1}{\varepsilon_m},\tag{3}$$

where $\varepsilon_m$ is the a priori volume transport uncertainty for each layer and additional constraints (e.g. FC transport, DWBC, etc.). For property conservation, the row weighting is defined as

$$W_{mm} = \frac{1}{2\eta_m^C \varepsilon_m},\tag{4}$$

where $\eta_m^C$ is the standard deviation of properties within layer $m$. To take into account a possible correlations between the section averaged and mesoscale components of the noise in the property conservation equations, a factor of 2 was introduced

in the denominator in the right hand side of Eq. 4 (Ganachaud and Wunsch, 2000; Tsubouchi et al., 2012). Following Ganachaud (2003), we set the uncertainties for volume conservation equations to gradually decrease from 8.2 Sv in the surface layer to 0.5 Sv in the bottom layer, and 15.9 Sv for the whole box.

The column weighting was set based on the a priori uncertainties of the reference velocity, $\delta b_j$, and the dianeutral velocity, $\delta w_m$. For the reference geostrophic velocity, the column weighting is

$$E_j = [\delta b_j / A_j]^{1/2},\tag{5}$$

and for the dianeutral velocity, the column weighting is

$$E_m = [\delta w_m / Q_m]^{1/2},\tag{6}$$

where $A_j$ and $Q_m$ are the vertical area between the station pairs and the horizontal area of the layer interfaces (neutral density surfaces), respectively. Except for the 2013 realizations at 14.5° N, a priori uncertainties for the reference velocity of the

other three realizations were assumed to be 0.04 m s$^{-1}$ at the western boundary region due to its large variability, and 0.02 m s$^{-1}$ for the rest of the station pairs (Ganachaud, 2003; Hernández-Guerra et al., 2014). For the 2013 realization, the a priori





uncertainties at the western boundary were estimated based on the variability of the SADCP velocity (Fig. 7, see Sect. 3.3 for details). The dianeutral exchanges of volume, salt and heat are expected to be small in the tropical Atlantic, we set the a priori uncertainty of the dianeutral velocity to be in the order of $10^{-6}$ m s$^{-1}$.

Applying the row and column weightings to Eq. 2, we have

$$(WAE)(E^{-1}x) = Wb, \tag{7}$$

which can be written as

$$A''x' = b', \tag{8}$$

with $A'' = WAE$, $x' = E^{-1}x$, and $b' = Wb$. The weighted system can be solved by using singular value decomposition (SVD) (Wunsch, 1996) by choosing a rank where data residual norms are of O[1 to 2 Sv] (Sloyan and Rintoul, 2001). An error covariance matrix $P$ can be formulated using the Gauss-Markov method. The a posteriori errors of the solution were estimated as the square root of the diagonal components of $P$, where

$$P = E - EA^T(AEA^T + W)^{-1}AE. \tag{9}$$

**3.5 Sensitivity tests of the inverse model**

It is of interest to test how sensitive the inverse results are to the initial conditions and constraints. The 1992 realization at 24.5° N have been used in several previous studies (e.g. Ganachaud, 2003; Lumpkin and Speer, 2003; Hernández-Guerra et al., 2014), we found that the circulation structure and strength is very stable among these studies, even though different hydrographic sections were combined with this realization to build the box inverse models. This can be attributed to the similar steady-state initial condition and constraints applied to the box inverse models. To further quantitatively answer the question, we performed three sensitivity experiments on the 14.5° N section as follows and referred to the box inverse model described in Sect. 3.1-3.4 as the "control run".

(1) Sensitivity to the meridional Ekman transport. In this experiment, we used the 1989/1992 box and artificially doubled the Ekman transport at 14.5° N to 17.6 Sv, everything else is identical to the control run.

(2) Sensitivity to the initial reference velocity. In the "control run" of the 2013/2015 box, the reference velocity for the 14.5° N section was estimated using the SADCP/LADCP velocity. In order to test how much the initial reference velocity affects the finial transport, we replaced the initial reference velocity at 14.5° N of the 2013/2015 box with 0, and kept everything else identical to the control run.

(3) Sensitivity to hydrographic variability. To assess the effect of hydrographic variability on the inverse model solution, we conducted 6 experimental runs using the 2013 realizations at the 14.5° N section in combination with six realizations of the 24.5° N section (1992, 1998, 2004, 2010, 2011, and 2015, data available from CCHDO and World Ocean Database). For all six experimental runs, we used the long-term mean Ekman transport calculated using the monthly NCEP/CFSR wind stress (1979-2010) for both sections and a long-term mean FC transport (31.1±2.4 Sv) at 24.5° N, and kept everything else identical to the control run. In this way, we could examine the stability of the results at 14.5° N against the hydrographic variations at 24.5° N.



### 3.6 Heat and freshwater flux estimation

The heat and freshwater fluxes through the two sections were estimated using the results from the box inverse model. Note that after the inversion the total volume flux through a section should be quasi zero (except a 0.5-0.8 Sv Bering Strait transport). Therefore, the total heat flux through a section, $H_{sect}$, can be regarded as a sum of warm northward Ekman transport at the surface, $H_e$, and a much colder southward returning geostrophic flow over the full water column, $H_g$:

$$H_{sect} = H_e + H_g, \tag{10}$$

$H_e$ was calculated as

$$H_e = \frac{C_p}{f} \sum_{j=1}^{N} \theta_j^{surf} \tau_j^x \Delta x_j, \tag{11}$$

where $f$ is the Coriolis parameter; $C_p$ is the specific heat capacity of seawater; $\theta_j^{surf}$ is the mean in-situ sea surface potential temperature (SST) between the station pair; $\tau_j^x$ is the zonal wind stress from NCEP/CFSR. In this study, we use $g$ and $C_p$ as constants of 9.8 m s$^{-2}$ and 4000 J kg$^{-1}$ K$^{-1}$, respectively. Note that we only use the in-situ SST to calculate the Ekman heat transport, since it has been shown by Fu et al. (2017) that in the tropical Atlantic the Ekman heat fluxes estimated using in-situ SST and temperature profile data within the Ekman layer only differ marginally. $H_g$ was calculated as

$$H_g = \rho C_p \iint \bar{\theta} v^{abs} \, dx dz = \rho C_p \sum_j \sum_m \bar{\theta}_{m,j} v_{m,j}^{abs} \Delta x_j \Delta h_{m,j} , \tag{12}$$

where $v_{m,j}^{abs}$ is the absolute geostrophic velocity in the cell of layer $m$, bounded by station pair $j$, as obtained from the box inverse model; $\bar{\theta}_{m,j}$ is the mean in-situ potential temperature in the same cell of $v_{m,j}^{abs}$; $\Delta h_{m,j}$ is the thickness of layer $m$ at station pair $j$.

Freshwater flux through a section was achieved in the sense of salt and mass conservation between the Bering Strait and the corresponding section. It can be expressed as follows (cf. Friedrichs and Hall, 1993, and McDonagh et al., 2015 for details):

$$F_{sect} = M_{net} - \frac{M_{BS}\overline{S_{BS}} + M_e\overline{S_e} + M_g\overline{S_g} + M_w\overline{S_w}}{<S>}, \tag{13}$$

where $M_{net}$ is a net barotropic flow through the section; $M_{BS} = 0.8$ Sv is the Bering Strait transport, and $\overline{S_{BS}} = 32.5$ is the mean salinity at the Bering Strait; $< S >$ is the section-areal mean salinity; $M_e$, $M_g$, and $M_w$ are the Ekman and interior geostrophic, and western boundary volume transport, respectively; $\overline{S_e}$, $\overline{S_g}$, and $\overline{S_w}$ are the Ekman, interior geostrophic, and western boundary transport-averaged-salinity. $\overline{S_e}$ and $\overline{S_g}$ are defined as follows, $\overline{S_w}$ is analogue to $\overline{S_g}$:

$$\overline{S_e} = \frac{\frac{1}{f\rho}\sum_j S_j^{surf}\tau_j^x \Delta x_j}{M_e}, \tag{14}$$

and

$$\overline{S_g} = \frac{\sum_j \sum_m \bar{s}_{m,j} v_{m,j}^{abs} \Delta x_j \Delta h_{m,j}}{M_g}, \tag{15}$$

where $\bar{s}_{m,j}$ is the mean in-situ potential temperature in the same cell of $v_{m,j}^{abs}$; $S_j^{surf}$ is the mean in-situ sea surface salinity (SSS) between the station pair $j$; $\tau_j^x$ is the zonal wind stress at station pair $j$; $v$ is the absolute meridional geostrophic



velocity; $f$ is the Coriolis parameter. Not the western boundary current region at 14.5° N is defined by the western most 6 station pairs, and in the upper 1000 m, while at 24.5° N it is the Florida Strait.

## 4. Inverse model results

### 4.1 Adjustments and final transport

The final adjustments to the reference velocity were achieved by solving the box inverse model built with the 14.5° N and 24.5° N sections for the two periods (Fig. 8). For both periods, the largest adjustments appear in the boundary regions, where the a priori uncertainty of the reference velocity is large. For the rest of the sections, the adjustments are small and not significantly different from zero. Similar to the previous inverse model studies, the a posteriori errors have similar magnitude of the a priori uncertainties of the unknowns (Ganachaud, 2003; Hernández-Guerra et al., 2014).

In general, the dianeutral velocity is not significantly different from zero, except in the bottom layer ($\gamma^n = 28.152$ kg m$^{-3}$) of the 2013/2015 period, where strong upward velocity is observed (Fig. 9). A similar structure of the dianeutral velocity was also achieved by Ganachaud (2003) for the tropical North Atlantic box bounded by 7.5° N and 24.5° N. The densest AABW flows northward in the tropical North Atlantic at the bottom of the western basin, it mixes with the LNADW, resulting in water mass transformation and volume reduction of the AABW. Therefore, it is expected that upwelling occurs in the bottom

layer. However, in the 1989/1992 period a downwelling velocity appears at $\gamma^n = 28.152$ kg m$^{-3}$ due to a divergence of the horizontal geostrophic transport in the bottom layer (Fig. 10a). Note that the horizontal area of the surface $\gamma^n = 28.152$ kg m$^{-3}$ is very small since it only exists in the abyssal basin west of the MAR. Therefore, a downwelling velocity in the bottom layer will not change the fact that the AABW as whole upwells, when the upwelling velocity and the much larger horizontal area of the layer surface $\gamma^n = 28.141$ kg m$^{-3}$ is taken into account.

Using the adjustments to the reference velocity and the dianeutral velocity, the zonally integrated final transport per layer is calculated, including the Ekman transport in the first layer (Fig. 10). Compared to the initial transport (not shown), clear overturning structures with southward NADW and northward upper, intermediate and bottom waters are achieved for both sections during the respective surveys. The imbalance in layers has been reduced indistinguishable from zero, which implies volume conservation in both boxes after inversion. The final transport of water masses can be then calculated by integrating

the layer transport over the corresponding layers of a water mass. As summarized in Table 3, for the 14.5° N section, the thermocline water (layer 1-3) has northward transport of 5.0±1.7 Sv in 1989, and 3.7±3.3 Sv in 2013. The intermediate water (layer 4-6) transport is 4.2±2.2 Sv northward in 1989 while only 2.6±1.7 Sv in 2013, which is considerably weaker in the latter period. Note that due to the higher weighting (small uncertainty in the constraints) of the intermediate water transport at 14.5° N in 2013, the final intermediate water transport in 2013 is very close to the prescribed value of 2.8±2.1 Sv. It is

important to point out that without this constraint, the final intermediate water transport would be even southward, no matter which reference level is chosen, and whether the SADCP/LADCP reference velocity is applied or not. This is undesirable and can be seen as a strong indication that the intermediate water transport in 2013 is substantially weaker than that in 1989.

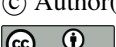



Recall the increasing temperatures and salinities in the AAIW layer at 14.5° N between 1989 and 2013, which may be explained by the reduction in the northward supply of fresh and cool AAIW water between the two periods.

In response to the much weaker northward thermocline and intermediate water transport, the UNADW (layer 7-11) transport of -6.8±2.6 Sv in 2013 is also much weaker than that of -12.1±3.2 Sv in 1989 ("-" denotes southward). The LNADW (layer

12-15) transport of the periods is very close to each other (-8.6±6.1 Sv in 1989 vs. -10.0±6.7 Sv in 2013). The AABW (layer 16-17) transport is very close between the two realizations, with 2.2±2.3 Sv in 1989, and 1.8±2.3 Sv in 2013, but the northward core of AABW transport shifts to a denser layer (from layer 16 to 17) between 1989 and 2013.

At 24.5° N, in contrast to the 14.5° N section, the circulation structure is relatively stable between 1992 and 2015 (Fig. 11b). The transport of thermocline water (layer 1-4) excluding the FC transport is -18.0±1.0 Sv in 1992, while it is -21.0±1.1 Sv in

2015, which is stronger southward. The northward intermediate water (layer 5-6) transport is also slightly weaker in 2015 (3.1±1.2 Sv) than in 1992 (3.5±1.2 Sv). A stronger southward transport in the thermocline water and a weaker northward transport in the intermediate water indicate together a weaker northward transport in the upper ocean in the more recent period than in the former period. As a result, the southward UNADW (layer 7-11) transport of -9.4±1.9 Sv in 2015 is also weaker than that of -12.2±1.7 Sv in 1992. The LNADW (layer 12-15) transport is indistinguishable between the two periods

(-9.9±5.2 Sv in 1992 vs. -10.4±5.5 Sv in 2015). The AABW transport is also nearly identical between the two periods (2.0±2.5 Sv in 1992 vs. 2.0±2.6 Sv in 2015).

In order to compare the intensity of the AMOC between the different periods, an overturning stream function of each realization was calculated by cumulatively integrating the layer transport from the surface to the bottom (Fig. 11). The AMOC strength is defined as the maximum northward transport of the overturning stream function. At 14.5° N, the AMOC

strength is 18.6±2.7 Sv in 1989, and 15.3±3.8 Sv in 2013. At 24.5° N, the AMOC strength is 19.2±1.5 Sv in 1992, and 17.7±1.6 Sv in 2015. It appears that at both latitudes the AMOC is weaker during 2013/2015 than during 1989/1992, although not significant. As described above, at both latitudes, we see a reduction of northward thermocline and intermediate water between the two periods, which is compensated by a decrease of the southward UNADW transport. It is worth to note that, the 1992 realization at 24.5° N has been used by many global and regional inverse studies (Ganachaud, 2003; Lumpkin

and Speer, 2003; Hernández-Guerra et al., 2014). Despite the fact that various other sections are combined with this realization to perform box inverse models, the vertical circulation structure and the transport strength are strikingly similar for this realization. All these studies have imposed a more or less time-mean Ekman transport and FC transport, as well as similar a priori errors (weightings) to the box inverse model. This indicates that given a steady-state initial condition, stable inverse results may be achieved, though the baroclinic structures and measurement errors (e.g. due to internal wave field) of

the combined section are very different.

### 4.2 Horizontal circulation pattern

The absolute geostrophic velocity sections are shown in Fig. 12. In the upper 1000 m at 14.5° N, intensified northward currents are confined at the western boundary. In 1989, the two branches of the western boundary current with nearly equal



amount of transport are located within a narrow channel and along the continental slope. A very weak southward flow separates the two branches in between. The western boundary current transport at 14.5° N in 1989, estimated by integrating the final total velocity from the western boundary to the eastern extent (58.30° W) of the northward current in the upper 1000 m, is 27.6±3.2 Sv, which is somewhat larger than an estimate of 18 to 24 Sv by Klein et al. (1995). However, the difference

does not come surprisingly, since their result was obtained by using a seasonal Ekman transport of 15.94 Sv, which is more than two times higher than an annual mean value of 7.9±3.5 Sv estimated by Fu et al. (2017). As Ekman transport and western boundary current are the dominant northward components of the AMOC in the tropics, such a strong Ekman transport would result in, to the lowest order of volume conservation, a weaker western boundary current. The impact of a large Ekman transport on the transport results is discussed as a case study later.

In 2013, the western boundary current was not evenly distributed into two branches; instead, it was much more confined in the narrow channel, and amounted to 26.9±3.7 Sv. This value compares favorably with a transport of 25.3±9.2 Sv estimated using the LADCP cross-section velocity and errors within the channel.  Along the continental slope it was much weaker, and almost replaced by a much stronger southward flow compared to that in 1989. At 24.5° N, the western boundary current system is primarily the FC with a transport of 31.2±0.3 Sv and 32.0±0.9 Sv in 1992 and 2015, respectively. These values are

not significantly different from the constrained values (cf. Table 2), and are consistent with a long-term mean value of 31.1±2.4 estimated by Atkinson et al. (2010).

Along the western boundary at greater depth, two southward velocity cores can be seen in all sections over a neutral density range of 27.82 to 28.072 kg m$^{-3}$ and 28.072 to 28.141 kg m$^{-3}$, respectively (Fig. 12), which are associated with the DWBC. The high oxygen concentration found in the deep western boundary region roughly coincides with the depth of the DWBC

velocity cores (Fig 5). For the 2013 realization at 14.5° N, the dual-core structure mentioned above is also evident in the oxygen section with oxygen concentration larger than 260 μmol kg$^{-1}$. The relatively lower oxygen concentration at about 3000 m between the two cores also coincides with a minimum southward flow in the corresponding neutral density layer (28.044< $\gamma_n$ <28.072 kg m$^{-3}$, Fig. 10b). This structure is consistent with the vertical structure of the NADW transport observed in the western basin by the MOVE at 16° N (Send et al., 2011). Send et al. (2011) showed that the annual mean

NADW transport has two southward maxima located at 2000 m and 4000 m, corresponding to the LSW and DSOW, respectively, and a minimum in between centered at 3000 m, corresponding to the ISOW. The high oxygen concentration in this area also implies the presence of relatively recently formed waters of northern origin (LSW, ISOW, and DSOW), which is also evident from higher transient tracer concentrations (e.g. Chlorofluorocarbon; Molinari et al., 1992).

In order to calculate the DWBC transport, an eastern boundary of the DWBC must be defined for integration purpose. This

boundary is defined by the eastward extension of the southward velocity in the NADW layers (27.82< $\gamma_n$ <28.141 kg m$^{-3}$) in the western basin. At 14.5° N, we used 51° W for both realizations as the eastern boundary of the DWBC, which coincides with the high-CFC cores measured during the 1989 realizations (cf. Fig. 3 in Molinari et al., 1992), and with the high-oxygen cores measured during the 2013 realizations (Fig. 5c), respectively. Note that this definition includes the northward recirculation branches near the continental slope. The resulting DWBC transport is -22.7±9.6 Sv in 1989 and -17.9±8.0 Sv in



2013, respectively. For the 24.5° N section, we used 72° W as the eastern boundary of the DWBC for both realizations, resulting in a DWBC transport of -26.8±4.6 Sv in 1992 and -25.4±4.0 Sv in 2015, respectively.

The DWBC transport estimates at 14.5° N compare favorably with an estimate of 25 Sv at 10° N in the Atlantic by Speer and McCartney (1991), and a mean DWBC transport of 26.0±8.0 Sv in the tropical North Atlantic by Molinari et al. (1992),

which is calculated by using repeated cruise sections between 14.5° N and the equator. Friedrichs and Hall (1993) have shown that at 8° N in the Atlantic, 27 Sv were transported southward within the DWBC along the continental slope, which was accompanied by a northward recirculation of 16 Sv primarily along the western flank of the MAR. They further schematically showed that the whole tropical North Atlantic adopted such a circulation pattern. However, in this study, at 14.5° N, northward recirculation occurred at the western flank of the MAR, but also near the continental slope to the east of

the southward cores of the DWBC. For the 1989 realization, the recirculation near the continental slope accounts to 12.5 Sv, while that at the western flank of MAR is 14.8 Sv. For the 2013 realization, the recirculation near the continental slope is 16.9 Sv, while northward velocity is found in a large area over the MAR, which is likely a compensating flow of the strong southward velocity with similar strength between 42 and 35° W (Fig. 12). Due to the limitation of the maximum operation depth (6000 m), the LADCP was not operated in this area. But a strong southward flow is still evident in the SADCP

sections in the upper 1200 m (not shown).

### 4.3 Sensitivity of the box inverse model

As described in Sect 3.5, three sensitivity experiments were conducted to test the sensitivity of the box inverse model to the different initial conditions and constraints. The results of the sensitivity tests are presented here and compared with the results presented in Sect. 4.1 and 4.2 (referred to as "Control run").

#### 4.3.1 Response to meridional Ekman transport

This experiment shows that the upper ocean geostrophic transport has an immediate response to the change in the Ekman transport at the same section. When the Ekman transport at 14.5° N in the 1989/92 box is artificially doubled (from 8.8 Sv in the control run to 17.6 Sv in the experimental run), the thermocline and intermediate water transport strongly decreases, from 5.0±1.7 Sv and 4.2±2.2 Sv (in the control run) to -0.6±1.7 Sv and 1.5±2.2 Sv, respectively. The NADW and AABW

transport as well as the overturning strength change only marginally. The transport at 24.5° N is also insensitive to the change in the Ekman transport at 14.5° N. This may explain the very small (even southward) AAIW transport (-0.7 Sv) in the seasonal case of Klein et al. (1995), who applied an extremely large Ekman transport (15.94 Sv) at 14.5° N to a box inverse model combining the 1989 realization at 14.5° N and 1957 realization at 8° N.

#### 4.3.2 Sensitivity to reference velocity

This experiment shows that the circulation pattern is sensitive to the reference velocity, but the AMOC strength is not. As shown in Fig. 7, the initial reference velocity (2013 at 14.5° N) along the continental slope (station pair 7 to 10) is mainly



southward. Removing the initial reference velocity of the 2013 realization at 14.5° N would decrease the UNADW transport from 6.8±2.6 Sv to 4.6±2.6 Sv, and the total DWBC transport from 17.9±8.0 Sv to 11.9±8.0 Sv, which is certainly too weak compared to the expected DWBC transport at this latitude (16-27 Sv, Molinari et al., 1992). Such a dramatic change in the DWBC is not surprising, provided the least-square nature of the box inverse method, it would always minimize the size of

the correction to the initial reference velocity (McIntosh and Rintoul, 1997). Therefore, the experimental run can hardly reproduce the magnitude of certain circulation elements (e.g. DWBC) defined by the reference velocity that is far away from 0. Besides, initializing the reference velocity using known (observed) velocity field is common in box inverse studies (e.g. Tsubouchi et al. 2012), and we believe it provides us a robust circulation pattern in this study.

### 4.3.3 Effect of hydrographic variability

The results show that the hydrographic variation at one section does affect the circulation structure and strength at the other section. Taking the AMOC strength as an indicator, in the six experimental runs combining the 2013 realization of 14.5° N, the AMOC strength at 14.5° N varies between 11.4±3.7 Sv (2013/2004 run) and 16.7±3.7 Sv (2013/1992 run) with a mean value of 14.4 Sv. These results are not significantly different from that of the control run (15.3±3.9 Sv). Since all the constraints and initial conditions were given using the long-term mean value and kept unchanged among the runs, the only

candidate for the change is the different baroclinic and horizontal shear structures that are related to interannual to decadal variability or/and the measured eddy and internal wave field. Note that among the six experimental runs, the AMOC strength at 14.5° N from the combinations of 2013/2010, 2013/2011, and 2013/2015 is very stable (15.6±3.7 Sv, 15.3±3.6 Sv, and 14.8±3.8 Sv, respectively). This is an indication of the importance to perform a box inverse model using sections occupied in the nearby years.

### 4.4 Heat and freshwater fluxes

At 14.5° N, we estimated the heat flux through the section as northward 1.03±0.14 PW (1 PW = 10^15 Watt) in 1989 and 1.11±0.15 PW in 2013. Despite the fact that the strength of AMOC is weaker in 2013 than in 1989, the heat flux in 2013 is even slightly larger than that in 1989. This can be attributed to the source of the transport changes and to the higher mixed layer temperature (a mean offset of 1.60 °C in the upper 50 m) during the occupation in May 2013 than in February 1989. As

shown in Fig. 11, the reduction of AMOC strength occurs mainly in the lower thermocline and intermediate water transport (layer 3-5), which carries much less heat northward compared with the surface water transport (layer 1-2). The surface water transport is even stronger in 2013 than in 1989. Compared with the estimates of 1.22 PW (annual case) or 1.37 PW (seasonal case) by Klein et al. (1995), our heat transport estimate for the 1989 realization is small. However, in the sensitivity test (Sect. 4.3.1), corresponding to a 17.6 Sv Ekman transport, the heat flux amounts up to 1.50±0.14 PW, which explains the

large heat fluxes in Klein et al. (1995). Fu et al. (2017) showed that in the tropical Atlantic the SST can be used to represent the transport-weighted mean temperature in the Ekman layer, and that the uncertainty of the Ekman volume transport





dominates the uncertainty of the Ekman heat flux. Combining their conclusion with the results of the sensitivity test to the Ekman transport, it is suggested that the total heat flux across 14.5° N is sensitive to the Ekman volume transport.

The heat flux through the 24.5° N section is estimated as northward 1.39±0.10 PW in 1992 and 1.08±0.11 PW in 2015. The strong decrease of the heat flux between the two years reflects the reduction of the northward transport in the surface layers

(layer 1-4, Fig. 11). The heat flux at 24.5° N has been estimated for several times in previous studies, especially for the 1992 realization; for instance, Ganachaud and Wunsch (2003) calculated a heat flux of 1.27±-0.15 PW in 1992; Hernández-Guerra et al. (2014) estimated the heat flux as 1.4±0.1 PW in 1992 and 1.2±0.1 PW in 2011. All the above listed heat flux estimates align with a RAPID array-based estimate of 1.25±0.36 PW (mean ± standard deviation) by Johns et al. (2011) at 26.5° N.

Freshwater flux across the 24.5° N section was -1.24±0.20 and -0.96±0.20 Sv in 1992 and 2015, respectively. For the 1992

realization, the estimate here agrees well with that of -1.23 Sv by Rosón et al. (2003) and -1.16 Sv by McDonagh et al. (2015). The estimate in 2015 falls in the range of -0.93 to -1.26 Sv based on hydrographic data or -1.17±0.20 Sv based on RAPID array data (see McDonagh et al. 2015 for a summary). At 14.5° N the freshwater flux was -1.00±0.15 and -1.02±0.22 Sv in 1989 and 2013, respectively. This is the first freshwater flux estimation at 14.5° N based on hydrographic data; compared to the estimates based on surface freshwater flux (i.e. integrating the evaporation, precipitation and river runoff

from a northern latitude with initial condition; Wijffels et al., 1992), it is somewhat larger. However, given a net surface freshwater loss of about 0.3 Sv between 24.5° N and 14.5° N estimated using HOAPS data and -1.17±0.20 Sv as the initial condition at 24.5° N, the estimates at 14.5° N are still within the uncertainty range.

Note that the uncertainties of the freshwater fluxes were estimated following Friedrichs and Hall (1993) as the amount of freshwater flux change due to an assumed salinity uncertainty of 0.2 psu in the western boundary at each section. This

assumption is based on the fact that the freshwater flux is very sensitive to the transport-weighted salinity in the western boundary ($\overline{S_w}$) rather than any transport component in Eq. 13. At both latitudes, an uncertainty of ±0.2 psu in $\overline{S_w}$ leads to a freshwater uncertainty of 0.1-0.2 Sv, while about 10 Sv change in $\overline{M_w}$ would be required to obtain a freshwater flux change of the same amount.

## 5 Comparison with the GECCO2 state estimate, RAPID and MOVE analysis data

The inverse results for the two sections during two different periods have shown overturning structure of comparable strength with previous studies in similar regions. At 24.5° N, the RAPID array data provide estimates of meridional transports across the section from Apr 2004 to Oct 2015 (Smeed et al., 2016). Here, we calculated the meridional overturning stream function using the absolute velocity field of the 2015 realization obtained from the inverse model, and compared it with the mean meridional overturning stream function of the RAPID array in 2015. Note that the transport for

the RAPID array is calculated based on depth levels. Therefore, for comparison, the overturning transport of the inverse results is also calculated as a function of depth. As shown in Fig. 13, the meridional overturning transport from the inverse method and the RAPID array are overall consistent with each other. The inverse result shows a slightly stronger overturning



strength (16.9±1.6 Sv for the inverse method, 16.1±4.4 Sv for the RAPID array), which is defined as the maximum transport in the meridional overturning stream function, typically at about 1000 m. The uncertainty of the RAPID array is given by the standard deviation of the RAPID time series (Apr 2004-Oct 2015). Note that the definition of water masses for the RAPID array is based on depth levels (thermocline water 0-800 m, intermediate water 800-1100 m, UNADW 1100-3000 m,

LNADW 3000-5000 m, AABW >5000 m), while for the inverse model, the water masses are defined by the neutral density surfaces. To best represent the depth definition of water masses, we have chosen the neutral density surfaces of 27.38, 27.83, 28.072, 28.1295 kg m$^{-3}$ as the boundaries between these water masses. Note that here the boundary between the LNADW and AABW is different from the previous definition only for comparison purpose. The thermocline water transport without the Florida current is -21.0±1.2 Sv for the inverse method, and -20.1±3.0 Sv for the RAPID array. The intermediate water

transport of the inverse method (1.6±1.2 Sv) is slightly stronger than the RAPID (0.6±0.6 Sv). The largest difference in the overturning stream function between the two estimates occurs at about 3000 m, the transport approaches to each other gradually between 3000 and 5000 m. This indicates that in comparison to the RAPID array, the inverse model underestimates the UNADW transport (-9.4±1.9 Sv for the inverse method, -11.1±2.5 Sv for RAPID), and overestimates the LNADW transport (-9.1±5.5 Sv vs -6.0±2.9 Sv), but the total transport of the NADW appears to be very close (-18.5±5.4 Sv

vs -17.1±4.0 Sv). The AABW transport is 0.7±2.6 Sv for the inverse method, and 1.0±0.6 Sv for RAPID array, also not significantly different from each other. This comparison shows that provided with time-averaged (annual mean) constraints, the box inverse model is able to reasonably reproduce the observed (RAPID) AMOC strength and structure.

In order to quantify the changes and variability of the AMOC with time, the AMOC strength was calculated using the results of the inverse method, and compared with that of the GECCO2 ocean state estimate (Jan 1985 to Dec 2014), the RAPID

array data (Apr 2004 to Oct 2015), and MOVE array data (Feb 2000 to Feb 2016) (Fig. 14). Köhl (2015) showed that the annual mean overturning stream function calculated from GECCO2 agreed very well with that from the RAPID array data in the upper 2500 m. In other words, GECCO2 represents the observed AMOC strength very well, as can be seen from the overlapping period between the assimilated (GECCO2) and observed (RAPID) AMOC strength time series (Fig. 14a). The overturning strengths of GECCO2 at both latitudes exhibit a strong seasonal cycle, with the maximum transport in boreal

winter, and minimum in boreal summer. The seasonal difference of the strength can reach up to 15 Sv. The 20-month low-passed AMOC strength of GECCO2 also reveals interannual variability at both latitudes. For instance, a continuous decline of the AMOC strength between 2008 and 2012 is captured by GECCO2 at both latitudes, which is also observed by the RAPID array. Smeed et al. (2014) analyzed different components of the AMOC (Ekman transport, western boundary current, mid-ocean geostrophic transport) at 26.5° N by using the RAPID array, and attributed the observed decline largely

to a strengthened southward transport in the upper mid-ocean (less than 1100 m). McCarthy et al. (2012) also pointed out that on the interannual time scale, the decrease of the AMOC strength was compensated by the strengthening of the horizontal gyre circulation. At the two latitudes, the AMOC strength calculated from the inverse method agrees with the GECCO2/RAPID AMOC strength in the respective years within error bars. The reduction of the AMOC strength in the inverse model between the two periods is also caused by a decrease in the upper ocean geostrophic transport.



The 120-day low pass filtered MOVE array transport time series at 16° N is shown in Fig 14b, as a representation and update of Fig. 2 in Send et al. (2011). The MOVE array is composed of a current meter mooring at the western boundary, and two end-point geostrophic moorings at each end of the western basin at 16° N, which measures the DWBC and interior geostrophic transport between 1200 and 4950 m in the western basin. An assumption behind this mooring configuration is

that the southward flow in the NADW layer in the western basin should represent (almost) entirely the southward return limb of the AMOC, thus the AMOC intensity (Send et al., 2011). Note that the sign of the MOVE transport in Fig. 14b is reversed for comparison purpose. In contrast to the good agreement between the AMOC strength calculations from the inverse method, GECCO2, and RAPID at 24.5° N, the AMOC strength from the MOVE array does not seem to align with that of the inverse method and GECCO2 at 14.5° N (Fig. 14b). Although the MOVE transport also shows seasonal to interannual

variability up to 20 Sv, the magnitude of the MOVE transport (24.1±4.1 Sv) is generally larger than that of GECCO2 at 14.5° N (15.8±3.4 Sv) throughout most of the overlapping period. Additionally, the MOVE transport shows a significant increase trend of 8.4±5.6 Sv decade[-1] since April 2004 (Baringer et al., 2015; by using data from Apr 2004 to Oct 2013 and 95% confidence interval), whilst the RAPID transport has a significant decrease trend of -4.1±3.2 Sv decade[-1] over the same period. Extending the data to date, the trend at 26° N reduces to -3.0 Sv decade[-1], while at 16° N it remains similar 8.2 Sv

decade[-1] (both are significant using two-sided t-test on 95% confidence interval). Since the NADW originates from the subpolar North Atlantic, and is transported southward, any long-term signal observed in the RAPID section should appear later in the MOVE section. Assuming the same transit speed as the water traveling from the Labrador Sea to 26° N (9 years, van Sebille et al., 2011), it would take 2-3 years for the same water to reach 16° N (Smeed et al., 2014). However, this does not seem to be the case when comparing results from this two observation systems. Speculations can be made to explain

these discrepancies, such as whether the assumption behind the MOVE array that the NADW through the western basin is representative of the whole southward return limb of the AMOC, or whether the reference level of no motion vary with time, etc., are fulfilled. The geostrophic velocity sections at 14.5° N (Fig. 12a, c) show northward recirculation over the MAR, which may not be fully resolved by the eastern most mooring of the MOVE array. However, it is impossible to verify whether it is a standing or transient feature based on the available data.

The AMOC strength derived from the GECCO2 state estimate does not show any trend at both latitudes over the presented period. Instead, the seasonal to interannual variability in the monthly and low-passed time series dominates the changes over different time periods. Therefore, we believe that the differences in the AMOC obtained from the inverse method between the respective realizations at the two latitudes are more likely due to the variability of the AMOC, rather than a long-term weakening or strengthening trend of the circulation.

**6 Conclusion**

In this study, hydrographic data from trans-Atlantic sections at 14.5° N occupied in 1989 and 2013, and at 24.5° N occupied in 1992 and 2015, are presented. Through comparison between the earlier realizations (1989, 1992) and the more recent





realizations (2013, 2015) at the respective latitude, we have shown property changes of water masses over time. At 14.5° N, basin-wide warming and salinification of AAIW on neutral surfaces is observed in the $\gamma_n$ range between 27.38 to 27.82 kg m$^{-3}$ (Fig. 3e and 4 e). Previous studies have also shown that at 7.5° N in the Atlantic, warming occurred in the intermediate layer (500-2000 m) through comparison of hydrographic data measured in 1957, 1993, and 2000 (Arhan et al., 1998;

Sarafanov et al., 2007, 2008), and a significant decrease of the northward AAIW transport at this latitude (Hernández-Guerra et al., 2014). Schmidtko and Johnson (2012) further showed warming and salinification trends of the AAIW core in the tropical North Atlantic since the mid-1970s. They attributed the trends in the Atlantic to an increase in winter SST in the AAIW formation region, and to a strengthened Agulhas leakage associated with a low SAM during some periods of the 20$^{th}$ century, which imports more warmer and saltier Indian Ocean water to the Atlantic (Beal et al., 2011).

The density of the densest AABW at both latitudes decreased over the studied period, at 14.5° N from 28.1686 kg m$^{-3}$ in 1989 to 28.1623 kg m$^{-3}$ in 2013, and at 24.5° N from 28.1596 kg m$^{-3}$ in 1992 to 28.1540 kg m$^{-3}$ in 2015. This implies that the denser AABW observed previously in the western basin has depleted, and that the lighter AABW type is now present, leading to a downward displacement of the isoneutrals. As a result, differencing $\theta$ on pressure levels, we observed an overall warming in the AABW layers with $\gamma_n$ larger than 28.141 kg m$^{-3}$ at 24.5° N, whilst at 14.5° N a warming at the bottom of the

continental slope (west of 55° W, and $\gamma_n$>28.141 kg m$^{-3}$). This is consistent with the previous studies by Johnson et al. (2008), who illustrated the warming of AABW in the North Atlantic using repeated zonal and meridional sections; and by Herrford et al. (2017), who showed that in the equatorial region, the temperature of the coldest AABW increased since the early 1990s.

By applying a box inverse model to the trans-Atlantic sections at 14.5° N and 24.5° N for a 1989/1992 period and a

2013/2015 period, the meridional overturning transport and the horizontal circulation pattern at the two latitudes and in different periods are obtained. The zonally integrated meridional transports per water mass for different realizations at the two latitudes are summarized in Table 3. Corresponding to the warming and salinification of the AAIW, we observed that at 14.5° N, the northward intermediate layer transport in 2013 was considerably weaker than in 1989 (Fig. 10 and 11a). We estimated that the western boundary current transport in 1989 amounted to 27.6±3.2 Sv, partitioned nearly equally into two

branches confined in the narrow channel and along the continental slope. In 2013, the western boundary transport amounted to 26.9±3.7 Sv, but was confined almost entirely within the channel, which is also supported by the LADCP data measured during the 2013 realization. Compared to an estimate of 18-24 Sv by Klein et al. (1995), our western boundary current in both realizations is strong, but we believe that their results are biased by an extremely large Ekman transport (15.94 Sv) used in their application of a box inverse model. The Lesser Antilles is located to the west of the 14.5° N section (Fig. 1),

connecting the Caribbean See with the Atlantic. Recent observation and model study suggest that a total flow of 17-19 Sv passes through the Lesser Antilles into the Caribbean Sea (Kirchner et al., 2008), implying that the western boundary current across the 14.5° N section must recirculate southward across the section in the east, as occurs in both realizations (Fig. 12a, b).



The DWBC transport at 14.5° N was -22.7±9.6 Sv in 1989 and -17.9±8.0 Sv in 2013, which is in agreement with the previous estimates ranging from -16 to -27 Sv in the tropical North Atlantic (Speer and McCartney, 1991; Molinari et al., 1992; Friedrichs and Hall, 1993). Different from Friedrichs and Hall (1993), we found that the northward recirculation occurred both at the western flank of the MAR, and directly to the east of the southward core along the continental slope. At

24.5° N, the DWBC transport was -26.8±4.6 Sv in 1992 and -25.4±4.1 Sv in 2015, which are ultimately close to the a priori constraint of -26.5±13.5 Sv, and agree with the estimates of -24.2 Sv, -26.5 Sv, and -32±16 Sv by Bryden et al. (2005), Johns et al. (2008), and Meinen et al. (2013), respectively.

The box inverse model results also show that at both latitudes the AMOC was weaker during 2013/2015 than during 1989/1992, which is consistent with the AMOC strength of the GECCO2 data at 14.5° N and 24.5° N during the

corresponding periods. However, long-term records of the AMOC in GECCO2 show strong seasonal to internannual variability, indicating that the observed decrease of AMOC intensity from the inverse method should be due to the variability rather than a long-term weakening trend of the AMOC. The variability of the AMOC may arise from different contributors on different time scales. Kanzow et al. (2010) concluded that at 26° N on subseasonal time scales the meridional Ekman transport is the main contributor, while on annual time scales the upper mid-ocean (upper 1100 m excluding the Florida

Strait) transport variability, due to seasonal changes of the wind stress curl near the eastern boundary, plays an dominant role. However, Frajka-Williams et al. (2016) analyzed the RAPID array data of 10-year record, and concluded that the upper mid-ocean variability was mostly counteracted by the FC transport on annual time scales, which, as a result, has little influence on the AMOC variability. Instead, it is the variability in the Ekman transport leaving an imprint on the AMOC variability on the same time scale.

The weaker AMOC at 14.5° N and 24.5° N during 2013/2015 in the inverse model was exhibited by a weaker northward thermocline and intermediate water transport, compensated by a correspondingly weaker southward UNDAW transport. Regarding the deep water formation as a primary driver of the AMOC, the long-term variability observed at lower latitudes might be related to changes in the export rate of deep waters in the subpolar North Atlantic. The variability of the NADW transport at 16° N estimated from the MOVE array was attributed to interannual variability in LSW transport (Send et al.,

2011). However, studies based on the RAPID array at 26° N show that the variability of the AMOC is more related to the changes in the LNADW transport rather than the UNADW (McCarthy et al., 2012; Smeed et al., 2014), contrary to the MOVE-based conclusion. The discrepancy in the source of the AMOC variability between the two latitudes is still unclear and needs further studies. Observations show that the LNADW source water transport, originating from the Nordic seas, is very stable (Hansen et al., 2016; Jochumsen et al., 2017). Observations also suggest that the formation of Labrador Sea

Water, which contribute to the UNADW, is not linked to the DWBC transport in the Labrador Sea (Zantopp et al., 2017). Consequently, variability in NADW formation is not expected to correlate well with the generation and propagation of transport signals in the DWBC.

We also estimated the heat flux through the 14.5° N section as 1.03±0.14 PW in 1989 and 1.11±0.15 PW. Compared to the estimates of 1.22 or 1.37 PW (annual or seasonal case) using the 1989 realization at 14.5° N by Klein et al. (1995), our



estimates are considerably smaller. This is mainly due to the anomalously strong Ekman transport applied in their version of box inverse model, as demonstrated in the sensitivity test of the box inverse model to the Ekman transport (cf. Sect. 4.3.1). At 24.5° N, the heat flux was 1.39±0.10 PW in 1992 and 1.12±0.11 PW in 2015, which are consistent with the previous estimates using hydrographic data ranging between 1.2 and 1.4 PW (e.g. Ganachaud and Wunsch, 2003; Hernández-Guerra

et al., 2014), and estimates based on the RAPID array of 1.25±0.36 PW (Johns et al., 2011).

As a closing note, we would like to raise some cautions when interpreting the box inverse model results, although we did not claim any long-term change of the AMOC at 14.5° N and 24.5° N. We argued that based on a number of box inverse studies, a time-averaged circulation can be achieved by applying time-averaged initial conditions to the box inverse model (Ganachaud and Wunsch, 2000; Ganachaud, 2003; Lumpkin and Speer, 2003; Hernández-Guerra et al., 2014).  However, it

is still not possible to rule out any potential influence of the temporal variability of the AMOC captured by the non-synoptic surveys on the inverse results; after all the two inverse boxes in this study were built on sections occupied in different seasons and different years. Intense seasonality of the equatorial current system and the tropical/subtropical cells would impact the western boundary current and the interior meridional transport. The interannual variability in the deep water formation region would also affect the returning limb of the AMOC. Through a sensitivity test, it is suggested that the

hydrographic variations at one section affect the circulation at the other section, and that it is maybe important to perform a box inverse model combining sections occupied in the nearby years (cf. Sect. 4.3.3 for details).

**7 Data availability**

The hydrographic data along 24.5° N in 1992, 1998, 2004, 2010, 2011, and 2015 are available at https://cchdo.ucsd.edu/. The NCEP/CFSR monthly wind stress data are available at http://rda.ucar.edu/datasets/ds093.2/. The NCEP/CFSv2 monthly

wind stress data are available at http://rda.ucar.edu/datasets/ds094.2/. The MOVE array data are available at http://mooring.ucsd.edu/index.html?/projects/move/move_intro.html. The RAPID array data are available at http://www.rapid.ac.uk/rapidmoc/rapid_data/transports.php. The Albatross IV data along the 14.5° N section are available at https://www.nodc.noaa.gov/OC5/WOD/pr_wod.html. The hydrographic and velocity measurements during cruise M09, M96 and M97 will be available through PANGEA at https://doi.org/10.1594/PANGAEA.870516.

**8 Acknowledgements**

This study is supported by the Deutsche Forschungsgemeinschaft as part of cooperative project FOR1740 and by European Union 7[th] Framework Programme (FP7 2007-2013) under grant agreement 603521 PREFACE project. We thank Thomas Müller for reprocessing the M09 CTD salinity, temperature, and oxygen data, and his comments on a previous version of the manuscript. We thank Toste Tanhua for closing the M96 section towards the African coast during M97. We thank Armin

Köhl for providing the GECCO2 data and the information about the data. Special thanks to Takamasa Tsubouchi for his





generous help on the box inverse model, as well as to Alonso Hernández-Guerra for his help on the error estimation of the inverse model results. The inverse model was developed from DOBOX version 4.2 (Morgan, 1994). Data from the RAPID-WATCH MOC monitoring project are funded by the Natural Environment Research Council and are freely available from www.rapid.ac.uk/rapidmoc. This work uses data from the MOVE project (see Kanzow et al., 2006, DOI:10.1016/j/dsr/2005.12.007). MOVE is a contribution to the international OceanSITES project.

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





**Table 1.** Neutral density surfaces ($\gamma_n$ kg m$^{-3}$) that separate the layers in the box inverse model. The water masses are labeled to the approximately corresponding layers.

| Layer | Lower surface ($\gamma_n$ kg m$^{-3}$) | Water mass |
|:---:|:---:|:---:|
| 1 | 26.44 | Surface water |
| 2 | 26.85 | |
| 3 | 27.162 | SACW/NACW |
| 4 | 27.38 | |
| 5 | 27.62 | |
| 6 | 27.82 | AAIW/MW |
| 7 | 27.922 | |
| 8 | 27.975 | |
| 9 | 28.008 | UNADW |
| 10 | 28.044 | |
| 11 | 28.072 | |
| 12 | 28.0986 | |
| 13 | 28.11 | LNADW |
| 14 | 28.1295 | |
| 15 | 28.141 | |
| 16 | 28.154 | AABW |
| 17 | Bottom | |





**Table 2. Constraints for the box inverse model and the final adjusted transport after inversion. Note that the box inverse model does not adjust the meridional Ekman transport, which is prescribed using the annual zonal wind stress during the calendar year of the cruise. The unit of the transport is Sv. Southward transport is given with "-".**

| | Position | Layers | 1989/1992 constraint | Final | 2013/2015 constraint | Final |
|---|---|---|---|---|---|---|
| **24.5° N** | | | | | | |
| **Ekman transport** | **Full section** | 1 | 4.5 | – | 4.6 | – |
| **Florida Current** | **Florida strait** | All | 31.3±1.1 | 31.2±0.3 | 31.7±1.1 | 31.8±0.9 |
| **Bering strait** | **Full section** | All | -0.8±0.6 | -0.9±6.3 | -0.8±0.6 | -0.8±6.6 |
| **DWBC** | **77:72° W** | 7:14 | -26.5±13.6 | -26.8±4.6 | -26.5±13.6 | -25.4±4.1 |
| **14.5° N** | | | | | | |
| **Ekman transport** | **Full section** | 1 | 8.8 | – | 8.3 | – |
| **Bering strait** | **Full section** | All | -0.5±0.6 | -0.5±7.8 | -0.5±0.6 | -0.5±8.3 |
| **Intermediate Water** | **Full section** | 4:6 | Not applied | 4.2±2.2 | 2.8±2.1 | 2.6±1.7 |



**Table 3. The final adjusted transport (Sv) of water masses, DWBC, and AMOC intensity at the two sections in the respective years. Southward transport is marked with "-". The values in the brackets at 24.5° N denote the transport results including the Florida current transport in the corresponding water mass layers.**

|  | 14.5° N | | 24.5° N (including Florida) | |
|---|---|---|---|---|
|  | 1989 | 2013 | 1992 | 2015 |
| Thermocline | 5.0±1.7 | 3.7±3.3 | -18.0±1.1 (11.2±1.1) | -21.0±1.2 (9.3±1.2) |
| Intermediate | 4.2±2.2 | 2.6±1.7 | 1.5±1.2 (3.5±1.2) | 1.6±1.2 (3.1±1.2) |
| UNADW | -12.1±3.2 | -6.8±2.6 | -12.2±1.7 | -9.4±1.9 |
| LNADW | -8.6±6.1 | -10.0±6.7 | -9.9±5.2 | -10.4±5.5 |
| AABW | 2.2±2.3 | 1.8±2.3 | 2.0±2.5 | 2.0±2.6 |
| DWBC | -22.7±9.6 | -17.9±8.0 | -26.8±4.6 | -25.4±4.1 |
| AMOC | 18.6±2.7 | 14.7±3.9 | 19.2±1.7 | 16.9±1.6 |

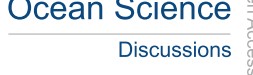



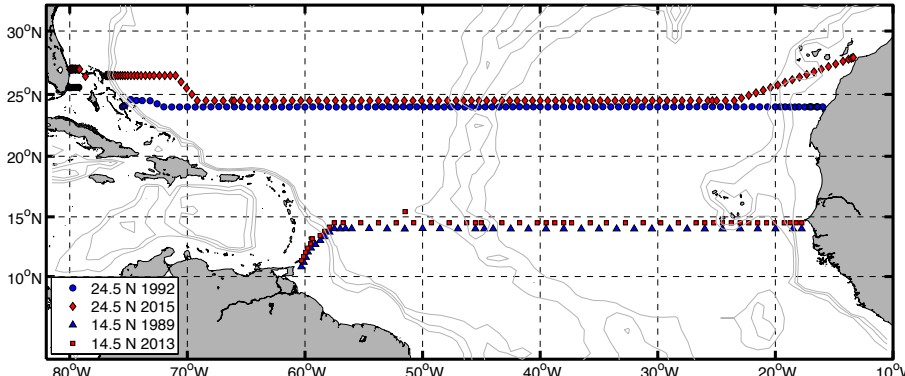

**Figure 1. Geographic map showing the hydrographic sections occupied in 1992 and 2015 at 24.5° N and in 1989 and 2013 at 14.5° N. Note that only the stations with CTD profiles reaching the bottom are shown here, and that the cruise tracks at both sections are offset by 0.5 degree latitude for visual clarity.**

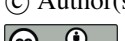



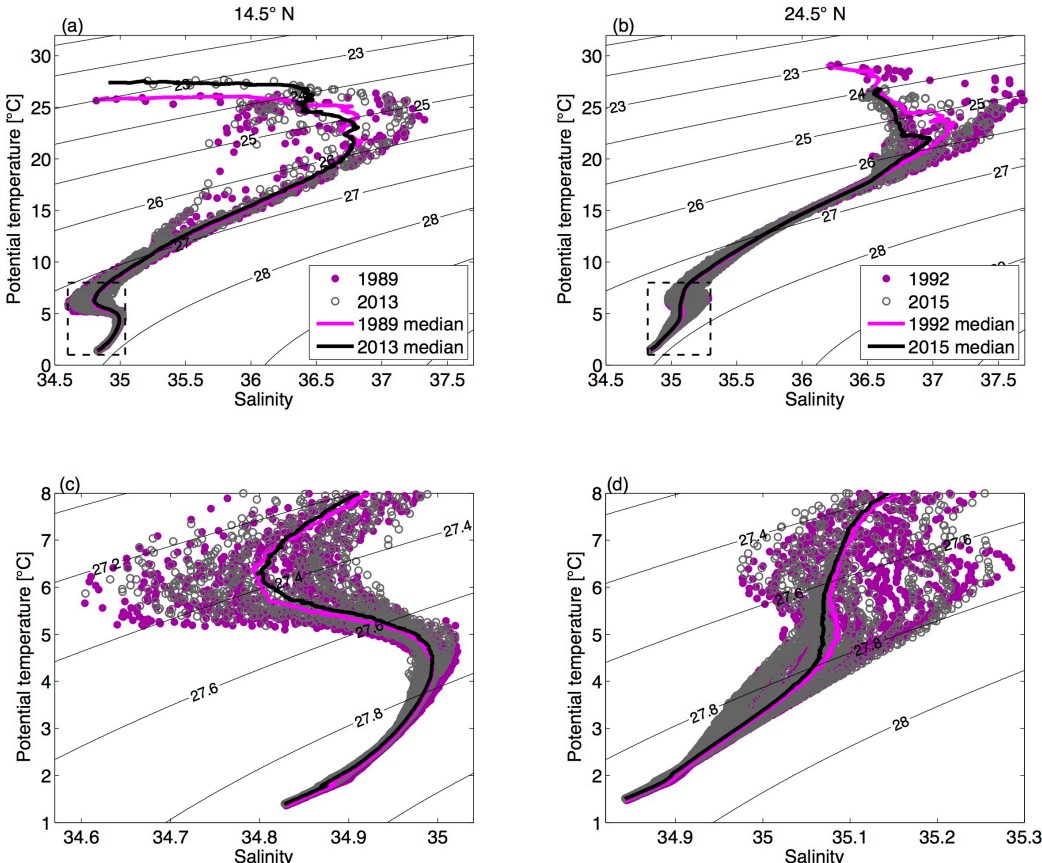

**Figure 2. The θ-S diagram at 14.5° N (a, c) in 1989 (dark violet solid circles) and 2013 (grey open circles), and at 24.5° N (b, d) for 1992 (dark violet solid circles) and 2015 (grey open circles). The median θ-S profiles, calculated on common neutral density grids, are shown in black for the earlier realizations at the respective sections and in light violet for the more recent realizations (cf. legend in a and b). The lower panel is the zoom of the area within the black dashed box in the corresponding upper panel. Note that the data are subsampled for visual clarity as every other profile horizontally and every 20 m vertically.**



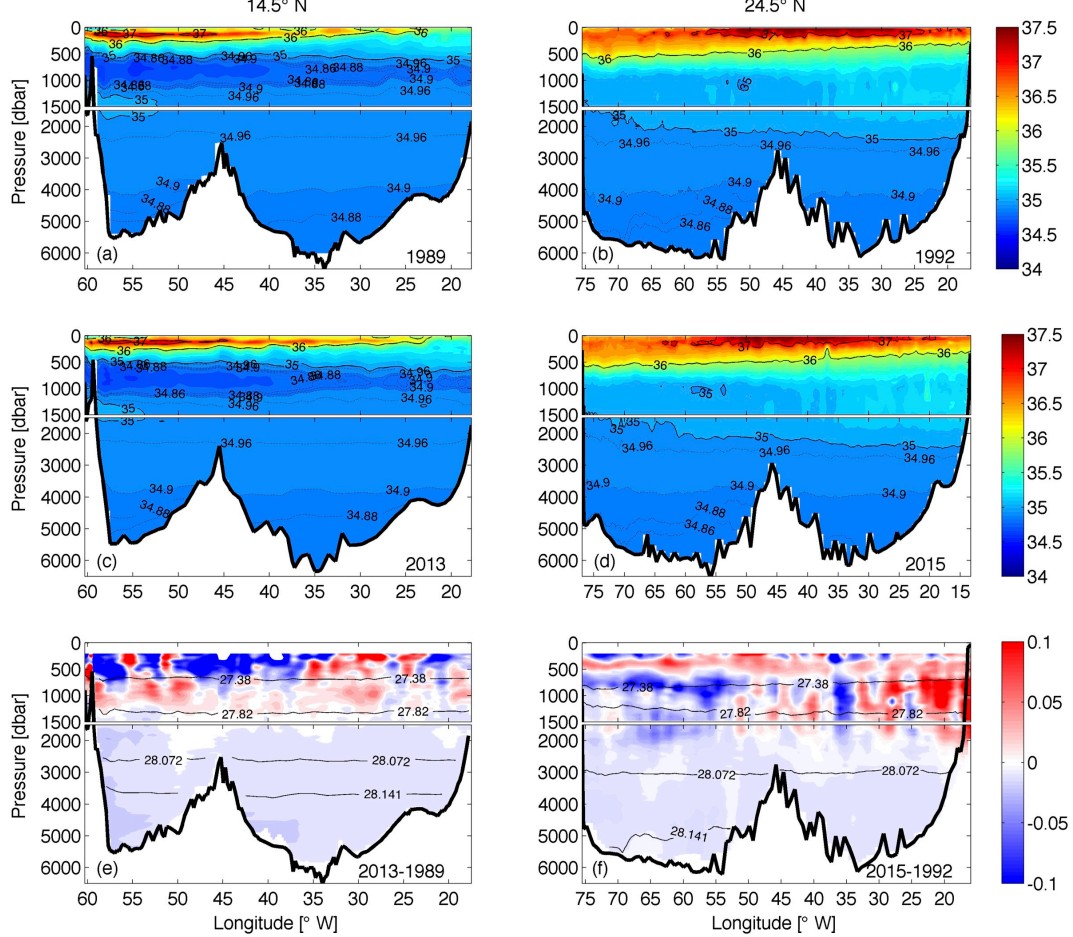

**Figure 3.** Vertical sections of salinity at 14.5° N in (a) 1989 and (c) 2013; and at 24.5° N in (b) 1992 and (d) 2015. Further shown is the salinity difference at (e) 14.5° N between 2013 and 1989, and (f) at 24.5° N between 2015 and 1992. Note that the difference is calculated first on neutral density levels, then transformed back onto depth levels, and that the upper 200 m are neglected. The contours in (e) and (f) are the neutral density surfaces of 27.38, 27.82, 28.072, and 28.141 kg m$^{-3}$, which marks the boundaries of AAIW, UNADW, LNADW, and AABW. The upper 1500 m in each subplot are stretched.





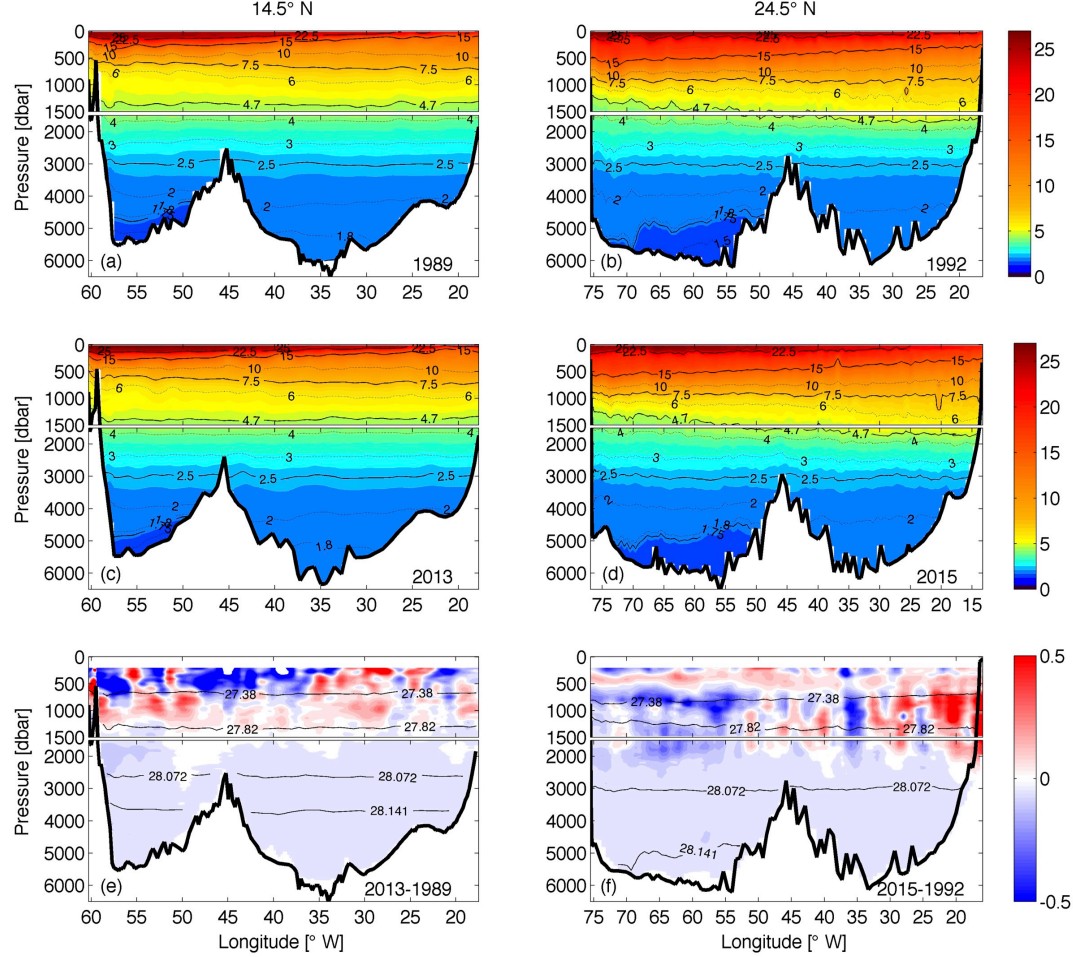

**Figure 4. Vertical sections of potential temperature (°C) at 14.5° N in (a) 1989 and (c) 2013; and at 24.5° N in (b) 1992, (d) 2015.**
5 **Further shown is the potential temperature difference at (e) 14.5° N between 2013 and 1989; and (f) 24.5° N between 2015 and 1992. Note that the difference is calculated first on neutral density levels, then transformed back to depth levels, and that the upper 200 m are neglected. The contours in (e) and (f) are the neutral density surfaces of 27.38, 27.82, 28.072, and 28.141 kg m$^{-3}$, which marks the boundaries of AAIW, UNADW, LNADW, and AABW. The upper 1500 m in each subplot are stretched.**





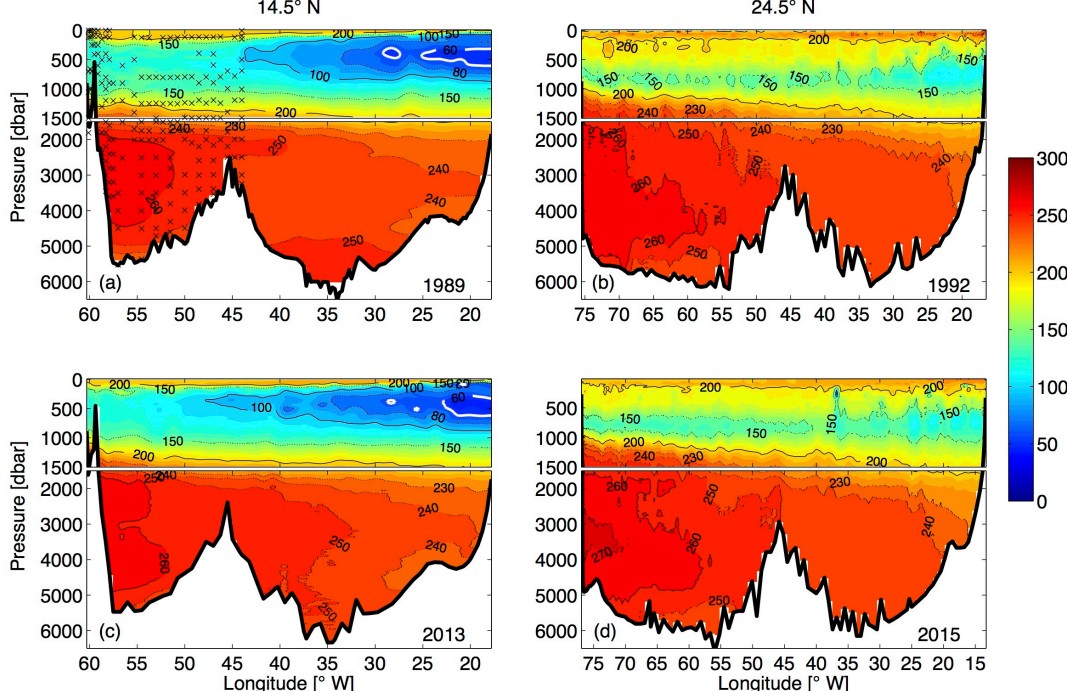

Figure 5. Vertical sections of oxygen (µmol kg$^{-1}$) at 14.5° N in (a) 1989 and (c) 2013; and at 24.5° N in (b) 1992 and (d) 2015. The "crosses" in (a) marks the position of bottle oxygen data, which are used to reconstruct the oxygen section in the western basin (see text for details). The white contour in (a) and (c) marks the 60 µmol kg$^{-1}$ oxygen. The upper 1500 m in each subplot are stretched.



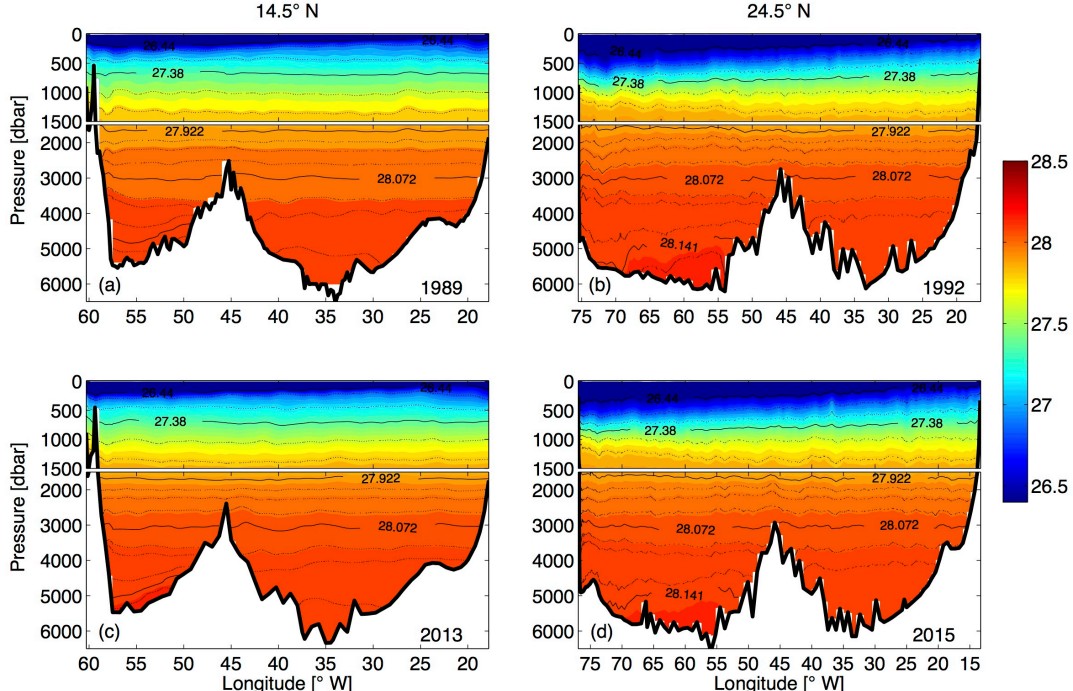

**Figure 6.** Vertical sections of neutral density (kg m$^{-3}$) at 14.5° N in (a) 1989 and (c) 2013; and at 24.5° N in (b) 1992 and (d) 2015. The upper 1500 m in each subplot are stretched.



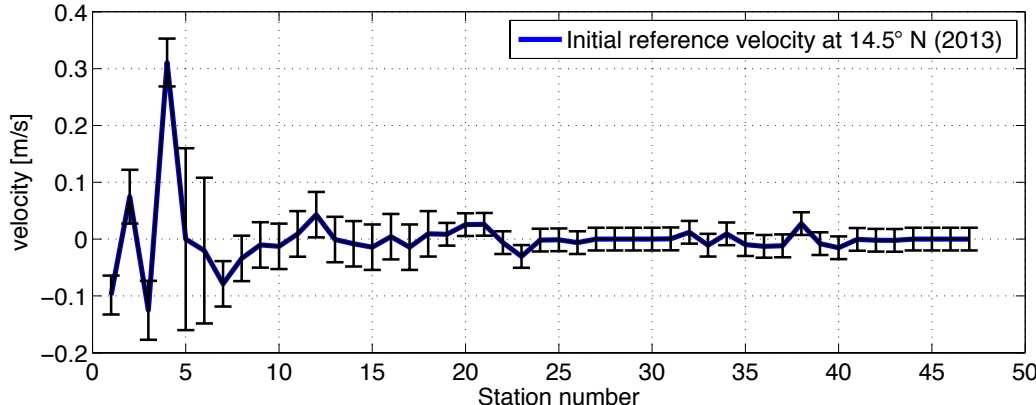

**Figure 7.** Initial reference velocity with error bars for the 2013 realization at 14.5° N, estimated from a combination of SADCP and LADCP (see text for details).





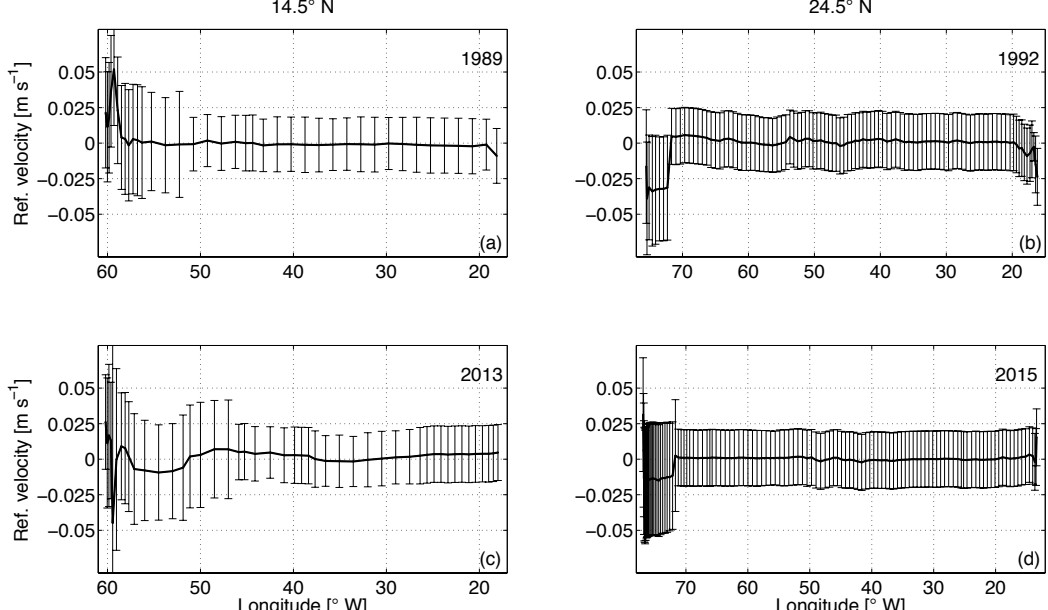

**Figure 8. Final adjustments to the reference velocity with uncertainties along 14.5° N in (a) 1989 and (c) 2013; and along 24.5° N in (b) 1992 and (d) 2015. Note that except for the section along 14.5° N in 2013, the initial reference velocity is 0; therefore, the final adjustments in (a), (b), and (d) are equivalent to the final reference velocity.**





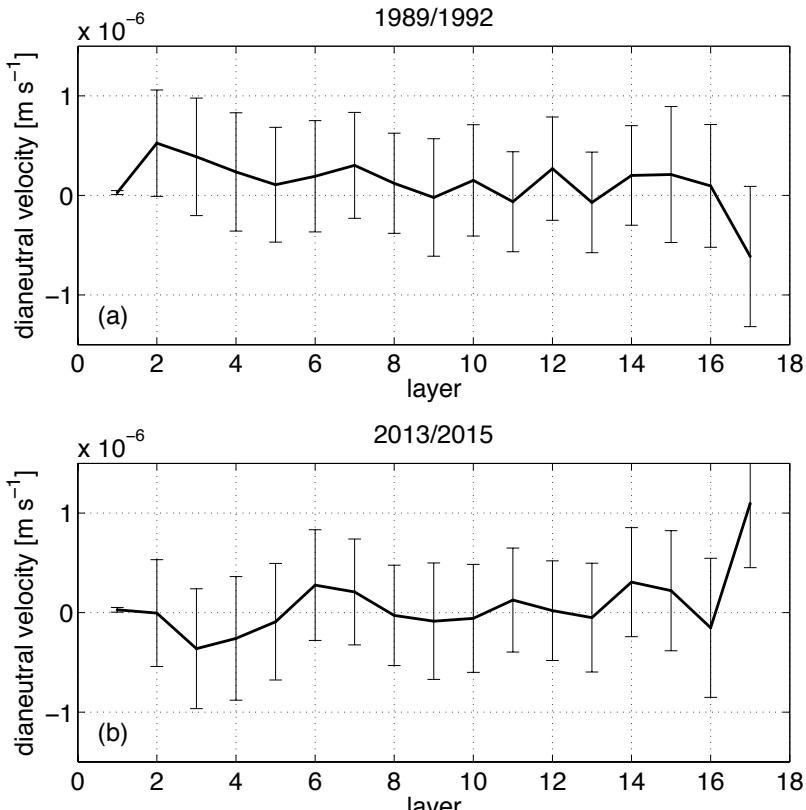

Figure 9. The dianeutral velocity across the layer interfaces between the two sections for the periods of (a) 1989/1992 and (b) 2013/2015.



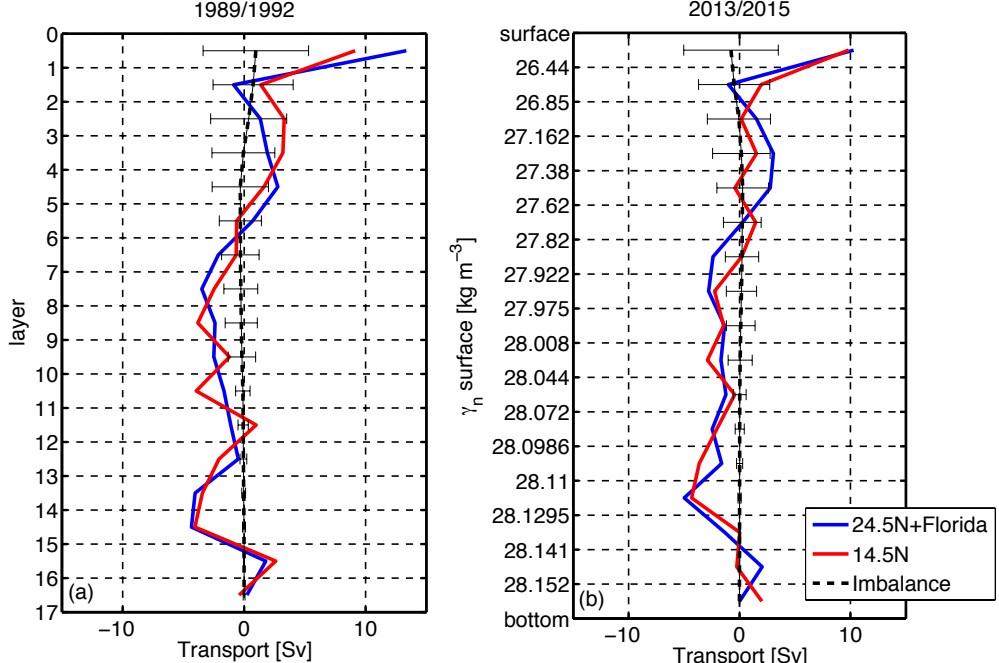

**Figure 10.** Final layer transports for the periods (a) 1989/1992 and (b) 2013/2015. The blue curves are the transports at the 24.5° N section including the Florida current transport, the red curves mark the transport at the 14.5° N section, the black dashed lines are the final imbalance of the layers with error bars.




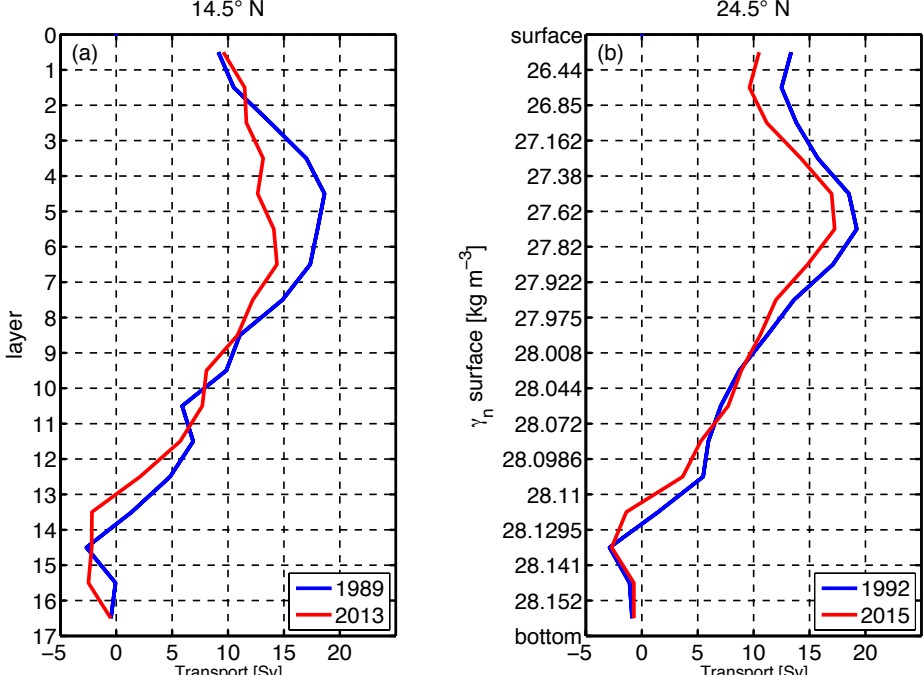

**Figure 11. Meridional overturning stream function from the inverse model at (a) 14.5° N and (b) 24.5° N. It is calculated by**
5 **cumulatively integrating the layer transport from the surface to the bottom. The vertical coordinate in (a) is the layer number, and in (b) the corresponding neutral density surfaces. For the 14.5° N section, the blue curve stands for the 1989 realization, the red curve for the 2013 realization. For the 24.5° N section, the blue curve stands for the 1992 realization, and red curve for the 2015 realization.**





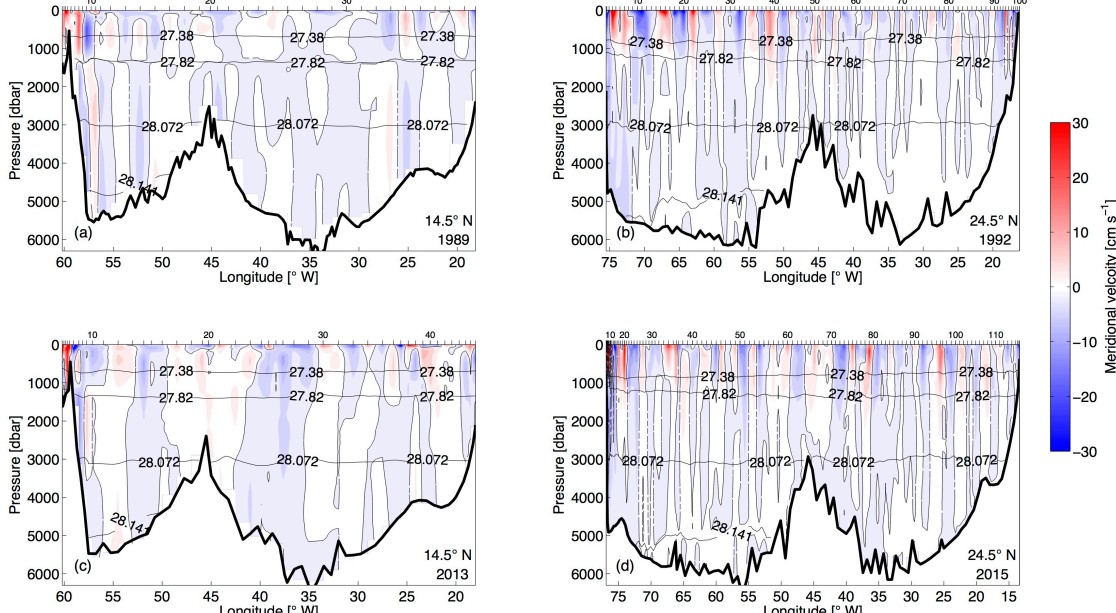

**Figure 12.** Absolute meridional geostrophic velocity along 14.5° N in (a) 1989 and (c) 2013, and along 24.5° N in (b) 1992 and (d) 2015. The station pair number is marked on the top horizontal axis. The black dashed contour line is the zero velocity contour. The black solid lines with values are the neutral density surfaces for the corresponding sections, which typically separate the thermocline water, AAIW, UNADW, LNADW, and AABW.





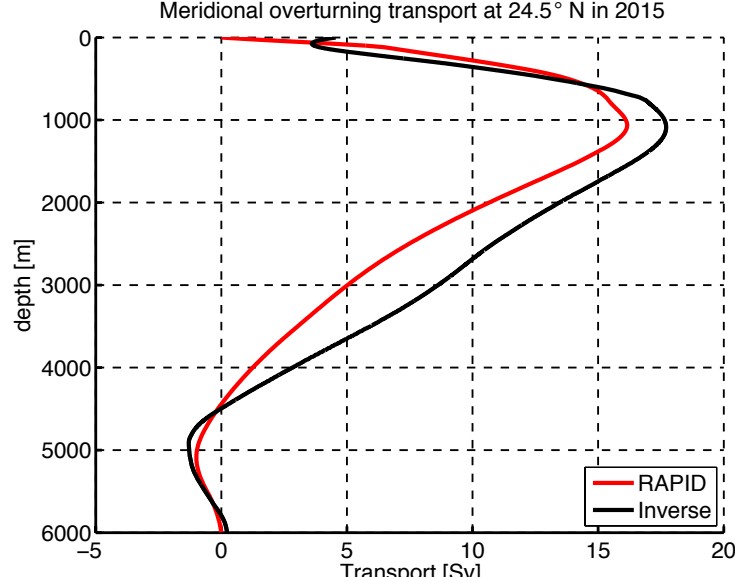

**Figure 13. Meridional overturning transport at 24.5° N in 2015, calculated from the inverse method (black) and RAPID array (red).**





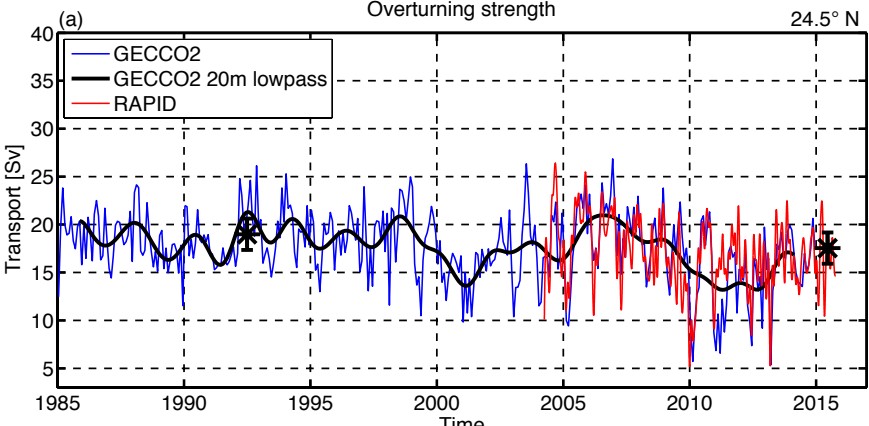

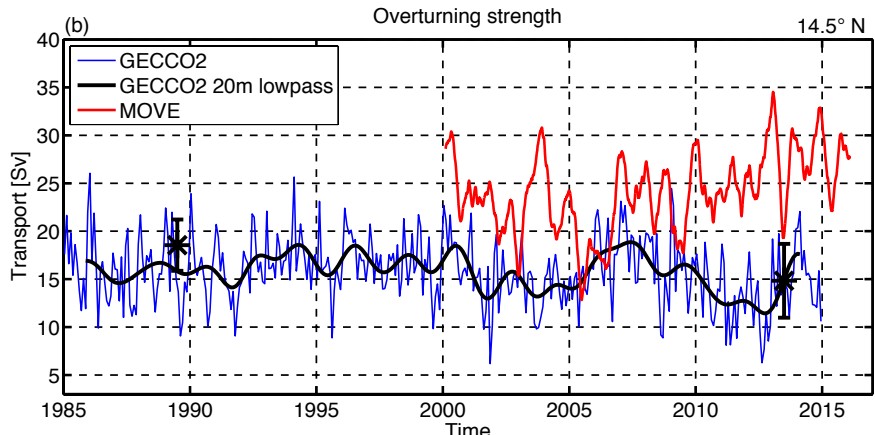

**Figure 14. Meridional overturning strength at (a) 24.5° N and (b) 14.5° N, calculated from the GECCO2 (blue), RAPID (red in a), MOVE (red in b), and inverse method (black stars with error bars for the respective years). The black thick line marks the 20-month low-passed time series of the GECCO2 meridional overturning strength. Note that the MOVE time series represents the southward transport in the NADW layer (1200 to 4950 m) in the western Atlantic basin, and that the sign of the MOVE time series is artificially reversed to ease the comparison.**