# Peer review of "Atlantic meridional overturning circulation at 14.5° N and 24.5° N during 1989/1992 and 2013/2015: volume, heat and freshwater transports"

_Ocean Science, 2017_

## Referee Comment (RC1) · Anonymous Referee #1 · 17 Dec 2017

I have looked at the work with thoroughness and consider that the authors have made an enormous effort to carry out a series of analyzes and generate significant knowledge about the Atlantic Meridional Overturning Circulation (AMOC). Despite these remarkable efforts, I can not recommend this article to be published in its current state. I believe that it is convenient to encourage the authors to review their work and introduce, if they deem it convenient, a series of modifications to make it more robust and ready to be finally published in your journal.

The authors essentially compare volume transports from four zonal hydrographic sections in the North Atlantic, performed with a time difference of 23-24 years, trying to

check if in that period the AMOC has intensified or weakened. To do this, they first analyze the distribution of water masses present in the study area and finally use an inverse model to estimate mass transport.

The main problem I see in the analysis presented is that the authors use for the 1989 section, made at 14.5°N, a reference level different than that used for the other sections. In particular, the reference level chosen is the neutral density ($\gamma$n) of 27.82 kg/m3, located approximately at 1500 m depth. For the other three sections, they place the reference level at a much greater value ($\gamma$n = 28.1295 kg/m3), located around 4000 m depth. The problem of placing a level of no-motion in the middle of the water column is that the values around that level of no-motion will also be close to zero, reducing notably the mass transports at those levels but allowing the values located further in depth to increase to unrealistically high values. It can be justified that different reference levels are used for each section, of course, but in this case in which the authors want to compare two sections made at different times, it would have been desirable for the reference level to be the same, so that the differences observed are due to the natural variability in the system and not to an artificiality introduced by the authors.

The inverse model is applied and it could introduce velocities in the reference level, but it actually doesn't. As it can be seen in the Fig. 8a, the reference level at the stations located in the interior ocean is practically 0. And in Fig. 10a, the no-motion level is clearly located at roughly 1500 m depth. So the initial choice of the reference level is relevant in the remainder of the paper, despite the usage of the inverse model.

If the authors insist on maintaining the current reference levels, it would be convenient for them to carry out some additional analysis in which they show that their results are not altered by the choice of the reference level. In particular, I would like to see what would have happened if the reference level of the section at 14.5°N of 1989 had been made with the deep reference level ($\gamma$n = 28.1295 kg/m3) or if the section at 14.5°N of 2013 would have been carried out with the shallow reference level ($\gamma$n = 27.82 kg/m3).

[Figure]

Comments in detail I would like to make some additional comments: 1. page 1, line 28. The authors' description of the formation of NADW is perhaps too simple, since not all the heat from the south goes to the formation of NADW but much of it is aimed at heating the climate of Europe. In any case, a reference is needed.

2. page 2, line 22. What type of changes? Changes in the formation rate? Changes in its thermohaline properties?

3. page 3, line 11. Why could it be problematic? Because of the different technology employed?

4. page 3. line 15. Please, before describing the structure of the paper, its main goal should be more explicitly indicated.

5. page 5, line 8. The authors describe a treatment performed on the dataset to interpolate and smooth the data for the sections at 14.5N. Could they provide a sensitivity analysis related to this treatment?

6. page 5, line 18. Could the authors also give more transparency to the analysis performed on the oxygen data?

7. page 6, line 33. There is a huge dispersion in the observations at those depths. Could the authors show if the differences plotted are significant?

8. page 7, line 8. At the eastern part of the North Atlantic, the gyre is not so deep to force the variability observed for the water masses at intermediate levels. The authors should consider rewriting this sentence.

9. page 9, line 1. $\Delta x_j$ is presented outside the integral. That would be right if the distance between stations would be regular, which is not the case. The authors should consider writing that element within the integral, as they did in Eq. 12.

10. page 9, line 15. The authors state that they have 51 unknowns. As far as I understood, the inverse model adds 17 unknowns as dianeutral velocities, which are

those later shown in Figure 9. The authors should consider rewriting this sentence.

11. page 10, line 5. The authors argue that they must modify meridionally the value of the transport at the Bering Strait (0.8 Sv) that has widely being used to constrain the net transports in zonal sections of the Atlantic Ocean. They state that, otherwise, they would be ignoring the freshwater flux from/to the atmosphere. So, if the freshwater flux between 14.5N and 24.5N is a positive value of 0.3 Sv to the atmosphere and the net balance at the zonal section at 24.5°N is 0.8 Sv, then the transport at 14.5°N should be 0.5 Sv. In their argument, the authors are somehow ignoring the freshwater flux in the basin north of the section at 24.5°N, which is unlikely 0. Not only Hernandez-Guerra et al. (2014) but also Ganachaud (2003) consider that the Bering Strait transport must be used in all the sections of the Atlantic Ocean. The authors should consider reviewing their approach in this point.

12. page 10, line 25. The sentence 'Given the much larger number of unknowns...' could be expanded with '...and the linear dependency between equations...'.

13. page 11, line 28. Please, consider producing a picture to explain the method to reference the velocities to those retrieved by the LADCP.

14. page 12, line 30. The authors should justify their choice for those uncertainties.

15. page 15, line 20. In Figure 10, the authors present the transports integrated along the layers. A third curve is given, which seems to be the imbalance between the other two curves. However, if we take a close look at the curves, we can see that the imbalance presented is a much smoother curve than the one that should appear after finding the difference between the other two curves. I would ask the authors why the imbalance curve is so smoothed, and why it doesn't seem to be related to the other two curves. In the left panel of this same figure, please, consider shifting the layer number so it is finally in the layer instead in its interface with the next layer.

16. page 16, line 5. The authors consider here that a difference of 1.4 Sv is '...very

close to each other', but in the previous page, on line 27, they consider that a difference of 1.6 Sv '...is considerably weaker'. They probably should rewrite those sentences.

17. page 17, line 12. I would suggest the authors to include a figure showing the velocities in the FC, as they did with the Figure 12.

A few typos 1. page 2, line 12: '...in detail...' 2. page 2, line 23: '... were occupied' 3. page 3, line 20: '... are given...' 4. page 3, line 24: please, expand here MAR. It's done later, on page 7, but this is the first time that MAR is used. 5. page 5, line 13: '(Figs. 3e-f and 4e-f)'. 6. page 7, line 4: please, expand SAM. 7. page 7, line 20: '...originated...' 8. page 15, line 1: 'Note the western...' 9. page 15, line 17: '...only exists...'
* * *

---

## Referee Comment (RC2) · Anonymous Referee #2 · 30 Jan 2018

General comments

This paper studies about changes of AMOC transports in four full depth basin wide hydrographic sections at 14.5N and 24.5N in 1989/1992 and 2013/2015. A box inverse model is used to derive the mass and salt conserved velocity fields. Authors find the quantified AMOC transports are stronger in 1989/1992 than 2013/2015 at both sections. They show the warmer and saltier signals in Antarctic Intermediate water (AAIW) at 14.5N sections in 2013 than 1989. The quantified four snapshots of the AMOC transports are compared against the RAPID array time series and a data assimilation product, GECCO2. They show the quantified transport differences between

1989/1992 and 2013/2015 are within range of seasonal and inter-annual variability. They suspect that these natural variabilities may explain the changes in the observed AMOC transports at 14.5N and 24.5N in 1989/1992 and 2013/2015.

Despite the large amount of efforts put into the study, I do not recommend to publish this manuscript in Ocean Science. My fundamental concern is, I do not see any clear scientific findings that advance our knowledge on the AMOC variability from the study. The time variability of the AMOC transports has been well studied based on the RAPID array at the 24.5N at different time scales, in terms of quantification and its driving factors: on seasonal time scale by Kanzow et al. (2010), on inter-annual time scale by McCarthy et al. (2012) and Frajka-Williams et al. (2016), and long-term trend by Smeed et al. (2012). In their figure 16, Kanzow et al. (2010) show that the aliasing of seasonal AMOC anomalies may be the major reason of the inferred slowdown of AMOC transports between 1957 and 2004 as proposed by Bryden et al (2005). Bryden et al. (2005) suggest the 8 Sv AMOC transport decline based on five full depth basin wide hydrographic surveys at 24.5N in 1957, 1981, 1992, 1998, 2004. These series of AMOC transport studies at 24.5N indicate two things. First, as far as I see, results in this study across 24.5N do not provide any new findings on the quantification of the AMOC transports. Authors assume zero reference velocities for both years in 1992 and 2015. This would hamper to quantify better snapshot estimates on AMOC transports than the RAPID mooring measurement. Second, given the lesson by Kanzow et al. (2010), comparing two snapshot estimates on AMOC transports across 14.5N in 10 years apart says little about the decadal AMOC transport change. As authors suspect, this change may be due to the natural seasonal and inter-annual variability. Again, I do not see clear scientific findings from the 14.5N sections.

References

Bryden, H. L., Longworth, H. R., and Cunningham, S. A.: Slowing of the Atlantic meridional overturning circulation at 25å̊Ůę N, Nature, 438, 655-657, 2005.

Frajka-Williams, E., Meinen, C. S., Johns, W. E., Smeed, D. A., Duchez, A., Lawrence, A. J., Cuthbertson, D. A., McCarthy, G. D., Bryden, H. L., Baringer, M. O., Moat, B. I. and Rayner, D.: Compensation between meridional flow components of the Atlantic MOC at 26° N, Ocean Sci., 12(2), 481-493, doi:10.5194/os-12-481-2016, 2016.

Kanzow, T., Cunningham, S. A., Johns, W. E., Hirschi, J. J.-M., Marotzke, J., Baringer, M. O., Meinen, C. S., Chidichimo, M. P., Atkinson, C., Beal, L. M., Bryden, H. L., and Collins, J.: Seasonal variability of the Atlantic meridional overturn- ing circulation at 26.5◦N, J. Climate, 23, 5678-5698, doi:10.1175/2010JCLI3389.1, 2010.

McCarthy, G., Frajka-Williams, E., Johns, W. E., Baringer, M. O., Meinen, C. S., Bryden, H. L., Rayner, D., Duchez, A., Roberts, C. D., and Cunningham, S. A.: Observed interannual variability of the Atlantic MOC at 26.5◦ N, Geophys. Res. Lett., 39, L19609, doi:10.1029/2012GL052933, 2012.

Smeed, D. A., McCarthy, G. D., Cunningham, S. A., Frajka- Williams, E., Rayner, D., Johns, W. E., Meinen, C. S., Baringer, M. O., Moat, B. I., Duchez, A., and Bryden, H. L.: Observed decline of the Atlantic meridional overturning circulation 2004- 2012, Ocean Sci., 10, 29-38, doi:10.5194/os-10-29-2014, 2014.

---

## Author Response (AR1)

**Author's response to the referee comments on the manuscript of "Atlantic meridional overturning circulation at 14.5° N and 24.5° N during 1989/1992 and 2013/2015: volume, heat and freshwater fluxes"**

**Yao Fu, Johannes Karstensen, and Peter Brandt**

5   yfu@geomar.de

We would like to thank both referees for their comments and suggestions. We found these very important and very helpful for improving the manuscript. Here are the author's responses to the comments. The referee's comment is first repeated in black color, followed by the author's response and the corresponding changes in the

10   manuscript in blue color. After the responses, we include a change-tracking version of the manuscript. The author's response part has page number in Roman numerals from I to XII; the change-tracking manuscript has page number in Arabic numerals from 1 to 65.

**Response to the anonymous Referee # 1**

I have looked at the work with thoroughness and consider that the authors have made an enormous effort to carry out a series of analyzes and generate significant knowledge about the Atlantic Meridional Overturning Circulation (AMOC). Despite these remarkable efforts, I cannot recommend this article to be published in its current state. I believe that it is convenient to encourage the authors to review their work and introduce, if they

20   deem it convenient, a series of modifications to make it more robust and ready to be finally published in your journal.

The authors essentially compare volume transports from four zonal hydrographic sections in the North Atlantic, performed with a time difference of 23-24 years, trying to check if in that period the AMOC has intensified or weakened. To do this, they first analyze the distribution of water masses present in the study area and finally use

25   an inverse model to estimate mass transport.

Thank you very much for the thorough examination and the following suggestions.

The main problem I see in the analysis presented is that the authors use for the 1989 section, made at 14.5° N, a

30   reference level different than that used for the other sections. In particular, the reference level chosen is the neutral density ($\gamma_n$) of 27.82 kg/m3, located approximately at 1500 m depth. For the other three sections, they place the reference level at a much greater value ($\gamma_n = 28.1295$ kg/m3), located around 4000 m depth. The problem of placing a level of no-motion in the middle of the water column is that the values around that level of no-motion will also be close to zero, reducing notably the mass transports at those levels but allowing the values

I

located further in depth to increase to unrealistically high values. It can be justified that different reference levels are used for each section, of course, but in this case in wrhich the authors want to compare two sections made at different times, it would have been desirable for the reference level to be the same, so that the differences observed are due to the natural variability in the system and not to an artificiality introduced by the authors.

5   The inverse model is applied and it could introduce velocities in the reference level, but it actually doesn't. As it can be seen in the Fig. 8a, the reference level at the stations located in the interior ocean is practically 0. And in Fig. 10a, the no-motion level is clearly located at roughly 1500 m depth. So the initial choice of the reference level is relevant in the remainder of the paper, despite the usage of the inverse model.

If the authors insist on maintaining the current reference levels, it would be convenient for them to carry out some
10  additional analysis in which they show that their results are not altered by the choice of the reference level. In particular, I would like to see what would have happened if the reference level of the section at 14.5° N of 1989 had been made with the deep reference level ($\gamma_n$ = 28.1295 kg/m3) or if the section at 14.5° N of 2013 would have been carried out with the shallow reference level ($\gamma_n$ = 27.82 kg/m3).

In order to show the impact of the reference level on the inverse solution, we conducted several sensitivity
15  experiments by using different reference levels for the 14.5° N section. The results are summarized in Table R1.

**Sensitivity experiments:**

- **Experiment 1,** deep reference level ($\gamma_n$ = 28.141 kg m$^{-3}$) for the 1989 realization

    In this experiment, we use the 1989/1992 box, and shift the reference level of the 1989 realization at 14.5° N from $\gamma_n$ = 27.82 to 28.141 kg m$^{-3}$, and keep everything else the same as described in the previous version of
20      the manuscript, except that we have used 0.8 Sv as the Bering Strait transport constraints for both sections.

- **Experiment 2,** shallow reference level ($\gamma_n$ = 27.82 kg m$^{-3}$) for the 2013 realization with initial reference velocity

    In this experiment, we use the 2013/2015 box, and shift the reference level of the 2013 realization from $\gamma_n$ = 28.141 to 27.82kg m$^{-3}$, then re-calculate the initial reference velocity using this reference level. Other settings
25      are kept the same as described in the previous version of the manuscript.

- **Experiment 3,** shallow reference level ($\gamma_n$ = 27.82 kg m$^{-3}$) for the 2013 realization **without** initial reference velocity

    In this experiment, except that the initial reference velocity is set to 0, other conditions are kept the same as in Experiment 2.

30  **Results:**

- **Experiment 1**

    After shifting the reference level of the 1989 realization at 14.5° N from $\gamma_n$ = 27.82 to 28.141 kg m$^{-3}$, we see that the thermocline and intermediate layer transports are altered very slightly. Given the same Ekman transport applied in the surface layer, this results in a nearly unchanged AMOC strength. However, the largest

difference occurs in the deep and bottom water layer, especially in the DWBC region. The DWBC transport changes from -21.5±9.6 (southward) to 6.5±9.6 (northward). The DWBC is defined as the transport between $\gamma_n = 27.82$ and 28.141 kg m$^{-3}$, west of 51° W. Although the total NADW transport does not decrease dramatically, this would mean that the main southward transport in the NADW layers occurs in the eastern half of the basin rather than within the DWBC. Additionally, the AABW transport also strongly decreased from 2.5±2.3 Sv to -0.3±2.3 Sv. These changes in the deep and bottom water layers are certainly not desirable. This is also the reason that we did not chose $\gamma_n = 28.141$ kg m$^{-3}$ as the reference level for the 1989 realization.

- **Experiment 2**

Here we use the 2013/2015 box and lifted the reference level of the 14.5° N section from $\gamma_n = 28.141$ to 27.82 kg m$^{-3}$. In this case, the initial reference velocity is calculated based on the new reference level, and applied. The result indicates that constrained by the initial reference velocity, the inverse solution would change only marginally by shifting the reference level for the 2013 realization at 14.5° N. The most obvious change is an increase in the DWBC transport by about 4 Sv. This can be expected since a shallower reference level would favor a stronger deep-water transport in the western boundary region, consistent with the results in Experiment 1.

- **Experiment 3**

In this experiment, we use the shallow reference level ($\gamma_n = 27.82$ kg m$^{-3}$) for the 14.5° N section of 2013 but set the reference velocity to zero. In this case, the inverse solution tends to return a weaker UNADW transport (-3.7±2.6 Sv), but a stronger LNADW transport (-13.2±6.6 Sv) in comparison to that in Experiment 2. The DWBC transport, as well as the AMOC strength in this experiment is stronger than in Experiment 2.

**Summary:**

For the 1989 realization of the 14.5° N section, the inverse solution is sensitive to the reference level, and using $\gamma_n = 27.82$ kg m$^{-3}$ as the reference level returns more robust features of the AMOC. For the 2013 realization, given the initial reference velocity as a constraint, the inverse solution is insensitive to the reference level, and both shallow and deep level returns consistent solutions. Removing the initial reference velocity would not significantly alter the AMOC strength but would change the structure in the NADW layers. This can be attributed to one of the limitations of the box inverse method. A box inverse model would always minimize the adjustments to the reference velocity to achieve volume conservation, and if a "true" value is too far away from the initial guess, a box inverse model would not be able to retrieve the "true" value without further constraints.

We agree with the Referee that in order to better compare the changes of the AMOC due to its own variability, the reference level for the 14.5° N section in the two periods should be kept the same. Based on the results of the sensitivity experiments, we decide to use $\gamma_n = 27.82$ kg m$^{-3}$ as the reference level for both realizations at 14.5° N, and constrain the 2013 realization with the initial reference velocity retrieved from the SADCP/LADCP. Additionally, we integrate the sensitivity experiments performed above in section 3.5 of the manuscript.

III

**Table R1 Transport results of the sensitivity experiments for the 14.5° N section. "Original" here means the inverse results from the previous version of the manuscript. For the 1989 realization, the original reference level is at $\gamma_n = 27.82$ kg m$^{-3}$; for the 2013 realization, the original reference level is at $\gamma_n = 27.1295$ kg m$^{-3}$.**

|  | 1989 | | | 2013 | | |
|---|---|---|---|---|---|---|
|  | Original | Exp 1 | | Original | Exp 2 | Exp 3 |
| Thermocline | 5.8±1.7 | 5.7±1.7 | | 2.8±3.3 | 2.9±4.8 | 4.7±3.2 |
| Intermediate | 2.9±2.2 | 3.1±2.2 | | 2.8±1.7 | 2.8±1.7 | 2.7±1.6 |
| UNADW | -13.2±3.2 | -9.6±3.2 | | -6.6±2.6 | -6.5±2.7 | -3.7±2.6 |
| LNADW | -7.5±6.1 | -8.4±6.1 | | -9.9±6.7 | -9.6±6.6 | -13.2±6.6 |
| AABW | 2.5±2.3 | -0.3±2.3 | | 2.0±2.3 | 1.4±2.4 | 0.5±2.3 |
| DWBC | -21.5±9.6 | 6.5±9.6 | | -17.4±8.0 | -21.3±8.0 | -23.4±8.0 |
| AMOC | 18.5±2.7 | 18.5±2.7 | | 14.1±3.9 | 14.3±5.3 | 16.2±3.8 |

Comments in detail I would like to make some additional comments:

1. page 1, line 28. The authors' description of the formation of NADW is perhaps too simple, since not all the heat from the south goes to the formation of NADW but much of it is aimed at heating the climate of Europe. In any case, a reference is needed.

Following this comment, we have changed this sentence to the following:

The warm upper-ocean water carries a large amount of heat from the tropics to the northern latitudes. It loses heat to the atmosphere on its way northward, loses buoyancy, and eventually sinks in the subpolar North Atlantic, returning southward as the cold NADW (Srokosz et al., 2012).

It is now in page 2 line 1-2.

2. page 2, line 22. What type of changes? Changes in the formation rate? Changes in its thermohaline properties?

It is the changes in the transport of LSW. The sentence has been modified as follows:

However, Send et al. (2011) found that at 16° N the interannual variability in the transport of the Labrador Sea Water (LSW; as the primary component of the upper NADW) is responsible for the interannual variability in the AMOC strength.

It is now in page 2, line 26-27.

IV

3. page 3, line 11. Why could it be problematic? Because of the different technology employed?

Because the hydrographic changes over 30 years may have an unknown impact on the inverse solution they obtained.

We therefore have added "due to hydrographic changes over the long period" at the end of the sentence.

5  It is now in page 4, line 21-22.

4. page 3. line 15. Please, before describing the structure of the paper, its main goal should be more explicitly indicated.

Following this suggestion, we have explained the main goal of this study and enriched the introduction
10  correspondingly.

5. page 5, line 8. The authors describe a treatment performed on the dataset to interpolate and smooth the data for the sections at 14.5N. Could they provide a sensitivity analysis related to this treatment?

The CTD measurement along the 14.5° N section was operated with an alternating depth between 3000 m and the
15  bottom. To better present the sections at 14.5° N, an interpolation and smoothing method was applied to these data. Please note that this method is used only to smooth and display the sections, neither for the property difference calculation, nor for the transport calculation. In the application of the box inverse model, only the profiles reaching the bottom were used.

A sensitivity analysis of this method is performed by using the salinity and potential temperature sections at 24.5°
20  N. Taking the salinity of the 1992 realization as an example, we first subsample the section by removing the salinity data below 3000 m in every other profile to mimic the case as at 14.5° N. Secondly, we interpolate and smooth the subsampled salinity on to the original grid using the same method as done to the 14.5° N section. Finally, a root mean square (rms) difference is calculated between the interpolated/smoothed salinity section and the original salinity section. The rms for salinity is 0.002 psu. The calculation of the rms for potential temperatur
25  is an analogue to the rms for salinity. The rms for potential temperature is 0.018 °C.

6. page 5, line 18. Could the authors also give more transparency to the analysis performed on the oxygen data?

In the western half of the basin at 14.5° N in 1989, we only have bottle oxygen data. In order to present a complete oxygen section, we need to apply a certain interpolation technique. We first tried to interpolate the
30  oxygen data in the vertical and zonal coordinates, but failed due to the very sparse sample position. We then applied a Gaussian weighting function to interpolate the oxygen data in the potential temperature and salinity coordinates. The assumption we made is that oxygen distribution should be related to water mass distribution, because water masses often have their own oxygen characteristics. This Gaussian weighting function is the same as used for interpolating the salinity and temperature data at the 14.5° N section. We first assigned the bottle

oxygen data to the CTD potential temperature and salinity data at the bottle positions, and then interpolated the oxygen data over the full range of all the CTD potential temperature and salinity data in the potential temperature and salinity space. Afterwards, the interpolated oxygen was gridded onto the CTD grid.

To further calibrate the interpolated oxygen, we then calculated a difference between the bottle oxygen and the
5    interpolated oxygen at the bottle positions. Since the bottle positions are scattered, we linearly interpolated the difference onto the CTD grid. Finally, we added this difference to the interpolated oxygen.

7. page 6, line 33. There is a huge dispersion in the observations at those depths. Could the authors show if the differences plotted are significant?

10   To show that the differences are significant, we calculated the standard error of the mean difference at each depth level, as shown in Fig. R1. The standard error at each depth is calculated as the standard deviation of the difference divided by the square root degrees of freedom (DOF) at each depth level. To calculate the DOF, we divide the total length of the section in the zonal direction by the integral length scale at each depth. The integral length scale is obtained by integrating the area between the first minus and plus zero-crossing of the
15   autocorrelation function of the difference at each depth along longitude.

The results show that the differences in the intermediate-water layer, deep-water layer, and bottom-water layer are significant.

[Figure]

Figure R1. Zonally averaged (a) salinity and (b) potential temperature differences at 14.5° N. The thick solid lines represent
20   the zonal mean differences, and the thin dashed lines represent the standard error of the mean in both plots.

8. page 7, line 8. At the eastern part of the North Atlantic, the gyre is not so deep to force the variability observed for the water masses at intermediate levels. The authors should consider rewriting this sentence.

This sentence has been rewritten as follows:

On the basin scale, a strengthening of the gyre scale circulation would bring more AAIW on the western side from lower latitude, while locally, for instance along the eastern margin, higher salinity in the 2015 realization than in the 1992 realization may reflect a seasonal intrusion of the less saline water near the African coast.

5   It is now in page 10, line 12-13.

9. page 9, line 1. $\Delta x_j$ is presented outside the integral. That would be right if the distance between stations would be regular, which is not the case. The authors should consider writing that element within the integral, as they did in Eq. 12.

10   This has been changed correspondingly.

10. page 9, line 15. The authors state that they have 51 unknowns. As far as I understood, the inverse model adds 17 unknowns as dianeutral velocities, which are those later shown in Figure 9. The authors should consider rewriting this sentence.

15   There are 17 unknowns for each property, but we have included dianeutral velocity across each neutral surface for volume, salt anomaly, and heat. As a result, there are 51 unknowns added to the system. Including dianeutral velocity for salt anomaly and heat in a box inverse model has been shown by e.g. McIntosh and Rintoul, 1997 as an effective way of parameterizing the net dianeutral fluxes.

20   11. page 10, line 5. The authors argue that they must modify meridionally the value of the transport at the Bering Strait (0.8 Sv) that has widely being used to constrain the net transports in zonal sections of the Atlantic Ocean. They state that, otherwise, they would be ignoring the freshwater flux from/to the atmosphere. So, if the freshwater flux between 14.5° N and 24.5° N is a positive value of 0.3 Sv to the atmosphere and the net balance at the zonal section at 24.5° N is 0.8 Sv, then the transport at 14.5° N should be 0.5 Sv. In their argument, the
25   authors are somehow ignoring the freshwater flux in the basin north of the section at 24.5° N, which is unlikely 0. Not only Hernandez-Guerra et al. (2014) but also Ganachaud (2003) consider that the Bering Strait transport must be used in all the sections of the Atlantic Ocean. The authors should consider reviewing their approach in this point.

We have followed this suggestion and changed the Bering Strait transport to 0.8 Sv for both sections in both
30   boxes, and further removed the surface freshwater flux constraint between the two sections. Since the magnitude of this adjustment is very small, it only changed the inverse results marginally. The text and transport values are modified correspondingly.

12. page 10, line 25. The sentence 'Given the much larger number of unknowns...' could be expanded with '...and

VII

the linear dependency between equations...'.

This has been modified accordingly.

13. page 11, line 28. Please, consider producing a picture to explain the method to reference the velocities to those retrieved by the LADCP.

To explain the method of retrieving the reference velocity from the LADCP, we would like to use one of the station pairs as an example. Here we use the station pair located between 25.25° W/14.5° N and 24.50° W/14.5° N, and produce Fig. R2. The red solid curve is the mean LADCP meridional velocity ($V_{LADCP}$) between this station pair; the red dashed line is the barotropic LADCP meridional velocity ($V_{LADCP\_baro}$), calculated by vertically averaging $V_{LADCP}$ between 200 m and 50 m above the last valid LADCP bin. Using this depth range is to avoid the influence of surface and bottom Ekman layer. The blue solid curve is the relative geostrophic velocity ($V_{geo\_rel}$) between this station pair; the blue dashed line is the mean relative geotrophic velocity ($V_{geo\_rel\_mean}$), calculated by vertically averaging $V_{geo\_rel}$ over the same depth range as done with $V_{LADCP\_baro}$. The initial guess of the reference geostrophic velocity for this station pair is then the difference between the two dashed lines ($V_{LADCP\_baro}$ - $V_{geo\_rel\_mean}$). Note that the barotropic tide component should be first removed from the LADCP velocity, which is not shown in this plot.

[Figure]

Figure R2. An example showing the relative geostrophic velocity profile and LADCP meridional velocity profile between the station pair located at 25.25° W and 24.50° W.

14. page 12, line 30. The authors should justify their choice for those uncertainties.

In general, if direct velocity observations are available, the uncertainties of the reference velocity can be estimated based on the observed variability, as we have done for the western boundary of the 14.5 N section in

VIII

2013. Since for most part of the sections, the variability of the reference velocity is unknown, the choice of the uncertainties is relatively arbitrary. On the one hand, we have assumed the energetic western boundary currents might have larger uncertainties than the currents in the interior ocean. On the other hand, larger uncertainties in the highly dynamical region means higher weighting in the box inverse model, which would give the reference velocity in western boundary region priority to be solved in the inverse model. We have followed Hernandez-Guerra et al. (2014) to assume an uncertainty at the western boundary of 0.04 cm s$^{-1}$, and 0.02 cm s$^{-1}$ in the interior ocean.

We have correspondingly modified the text.

15. page 15, line 20. In Figure 10, the authors present the transports integrated along the layers. A third curve is given, which seems to be the imbalance between the other two curves. However, if we take a close look at the curves, we can see that the imbalance presented is a much smoother curve than the one that should appear after finding the difference between the other two curves. I would ask the authors why the imbalance curve is so smoothed, and why it doesn't seem to be related to the other two curves. In the left panel of this same figure, please, consider shifting the layer number so it is finally in the layer instead in its interface with the next layer.

The imbalance after the inversion is relatively small and smooth compared to the transport difference in each layer between the two sections because this imbalance is calculated after taking into account the meridional and dianuetral transport in and out of each layer from the side and vertical boundaries. To make this point more obvious we now have clarified the imbalance in the caption of Fig. 10 and in the text.

We have now shifted the ticks of the left axis of Fig. 10 and 11 to the middle of each layer.

16. page 16, line 5. The authors consider here that a difference of 1.4 Sv is '...very close to each other', but in the previous page, on line 27, they consider that a difference of 1.6 Sv '...is considerably weaker'. They probably should rewrite those sentences.

We noticed this inconsistency and made corresponding modification.

It is now in page 23, line 11-12.

17. page 17, line 12. I would suggest the authors to include a figure showing the velocities in the FC, as they did with the Figure 12.

We have added a Figure 13 showing the meridional geostrophic velocity in the Florida Straits for 1992 and 2015. And correspondingly modify the text describing the Florida Current.

A few typos

1. page 2, line 12: '...in detail...'

Changed.

2. page 2, line 23: '... were occupied'

IX

We have modified the text in this part, and the original sentence containing this phrase has been deleted.

3. page 3, line 20: '... are given...'

Changed.

4. page 3, line 24: please, expand here MAR. It's done later, on page 7, but this is the first time that MAR is used.

It is expanded.

5. page 5, line 13: '(Figs. 3e-f and 4e-f)'.

Changed.

6. page 7, line 4: please, expand SAM.

Done.

7. page 7, line 20: '...originated...'

Changed.

8. page 15, line 1: 'Note the western...'

Changed.

9. page 15, line 17: '...only exists...'

Changed.

X

**Response to the anonymous referee #2**

This paper studies about changes of AMOC transports in four full depth basin wide hydrographic sections at 14.5N and 24.5N in 1989/1992 and 2013/2015. A box inverse model is used to derive the mass and salt
5  conserved velocity fields. Authors find the quantified AMOC transports are stronger in 1989/1992 than 2013/2015 at both sections. They show the warmer and saltier signals in Antarctic Intermediate water (AAIW) at 14.5N sections in 2013 than 1989. The quantified four snapshots of the AMOC transports are compared against the RAPID array time series and a data assimilation product, GECCO2. They show the quantified transport differences between 1989/1992 and 2013/2015 are within range of seasonal and inter-annual variability. They
10  suspect that these natural variabilities may explain the changes in the observed AMOC transports at 14.5N and 24.5N in 1989/1992 and 2013/2015.

Despite the large amount of efforts put into the study, I do not recommend to publish this manuscript in Ocean Science. My fundamental concern is, I do not see any clear scientific findings that advance our knowledge on the AMOC variability from the study. The time variability of the AMOC transports has been well studied based on
15  the RAPID array at the 24.5N at different time scales, in terms of quantification and its driving factors: on seasonal time scale by Kanzow et al. (2010), on inter-annual time scale by McCarthy et al. (2012) and Frajka-Williams et al. (2016), and long-term trend by Smeed et al. (2012). In their figure 16, Kanzow et al. (2010) show that the aliasing of seasonal AMOC anomalies may be the major reason of the inferred slowdown of AMOC transports between 1957 and 2004 as proposed by Bryden et al (2005). Bryden et al. (2005) suggest the 8 Sv
20  AMOC transport decline based on five full depth basin wide hydrographic surveys at 24.5N in 1957, 1981, 1992, 1998, 2004. These series of AMOC transport studies at 24.5N indicate two things. First, as far as I see, results in this study across 24.5N do not provide any new findings on the quantification of the AMOC transports. Authors assume zero reference velocities for both years in 1992 and 2015. This would hamper to quantify better snapshot estimates on AMOC transports than the RAPID mooring measurement. Second, given the lesson by Kanzow et
25  al. (2010), comparing two snapshot estimates on AMOC transports across 14.5N in 10 years apart says little about the decadal AMOC transport change. As authors suspect, this change may be due to the natural seasonal and inter-annual variability. Again, I do not see clear scientific findings from the 14.5N sections.

Thank you very much for your comments, which were most helpful in reconsidering our way of presenting our
30  results and sharping the objectives of this study. We understand your concern, and correspondingly modified the manuscript in a way that expresses our aim better. There are big differences between time series of AMOC strength from end-point geostrophic arrays such as RAPID, OSNAP, SAMOC and repeated ship sections such as OVIDE, 24.5° N, 14.5° N and more general "GO-SHIP sections". The sections are to seek to combine hydrographic/tracer changes and velocity structure changes while the end-point arrays are designed for time
35  series of fluctuations in mass transport. Here we concerned repeated hydrographic sections and examined water mass property changes at the two latitudes (14.5° N and 24.5° N) and for the past two decades. We have shown that the AAIW at 14.5° N became warmer and more saline. 14.5° N is probably the northernmost latitude, where AAIW property changes were observed. The NADW became fresher, while AABW became lighter at both

XI

latitudes. These results are in agreement with other observations at other locations. For both 14.5° N and 24.5° N sections, we used the newest available realizations (in 2013 and 2015, respectively), which may update our knowledge on water mass property changes in the tropical North Atlantic.

We would like to point out that this work does not intend to study the variability of the AMOC. We fully agree with reviewer #2 that only 4 snapshots are far from enough to examine the variability of the AMOC. We used the AMOC time series from GECCO2, RAPID and MOVE, but only to show that the inverse solution, even with the uncertainties that come with the solution, does fit the time series.

We hope that this "strategy" is no much clearer and invite this reviewer to look at the revised version of our manuscript. Thank you very much!

XII

[revised manuscript text omitted]